EMBO
Molecular Medicine

# Therapeutic implications of altered cholesterol homeostasis mediated by loss of CYP46A1 in human glioblastoma

Mingzhi Han[1,2,†], Shuai Wang[1,†] (iD), Ning Yang[1], Xu Wang[1], Wenbo Zhao[1], Halala Sdik Saed[2], Thomas Daubon[3,4], Bin Huang[1], Anjing Chen[1,5], Gang Li[1], Hrvoje Miletic[2,6], Frits Thorsen[2,7] (iD), Rolf Bjerkvig[2,8,*,‡] (iD), Xingang Li[1,‡,**] (iD) & Jian Wang[1,2,‡,***] (iD)

## Abstract

Dysregulated cholesterol metabolism is a hallmark of many cancers, including glioblastoma (GBM), but its role in disease progression is not well understood. Here, we identified cholesterol 24-hydroxylase (CYP46A1), a brain-specific enzyme responsible for the elimination of cholesterol through the conversion of cholesterol into 24(S)-hydroxycholesterol (24OHC), as one of the most dramatically dysregulated cholesterol metabolism genes in GBM. CYP46A1 was significantly decreased in GBM samples compared with normal brain tissue. A reduction in CYP46A1 expression was associated with increasing tumour grade and poor prognosis in human gliomas. Ectopic expression of CYP46A1 suppressed cell proliferation and *in vivo* tumour growth by increasing 24OHC levels. RNA-seq revealed that treatment of GBM cells with 24OHC suppressed tumour growth through regulation of LXR and SREBP signalling. Efavirenz, an activator of CYP46A1 that is known to penetrate the blood–brain barrier, inhibited GBM growth *in vivo*. Our findings demonstrate that CYP46A1 is a critical regulator of cellular cholesterol in GBM and that the CYP46A1/24OHC axis is a potential therapeutic target.

**Keywords** 24OHC; cholesterol homeostasis; CYP46A1; efavirenz; glioblastoma
**Subject Categories** Cancer; Pharmacology & Drug Discovery; Neuroscience

## Introduction

Glioblastoma (GBM) is the most common primary malignant brain tumour in adults (Fack *et al*, 2017; Lhomond *et al*, 2018). Despite aggressive therapy, including maximal surgical resection followed by radiotherapy and temozolomide treatment, the median patient survival is 14.6 months from initial diagnosis (Stupp *et al*, 2005). New treatment strategies are therefore urgently needed.

Emerging evidence has linked disrupted cholesterol homeostasis to cancer development including GBM (Bovenga *et al*, 2015; Kuzu *et al*, 2016; Cheng *et al*, 2018). However, epidemiological data remain contradictory regarding the relationship between cancer risk and serum cholesterol levels, suggesting that circulating cholesterol levels alone have marginal effects on cancer development (Silvente-Poirot *et al*, 2018). Oxysterols are oxygenated derivatives of cholesterol that participate in the regulation of cholesterol metabolism. They are derived from the diet or generated through endogenous cholesterol metabolism in tissues. Oxysterols include 24-hydroxycholesterol (24OHC), 25-hydroxycholesterol (25OHC), 27-hydroxycholesterol (27OHC) and ring oxysterols (Janowski *et al*, 1996; Berrodin *et al*, 2010) that are known to, in addition to cholesterol metabolism regulation, modulate signalling pathways such as Hedgehog, Wnt and MAPK (Kloudova *et al*, 2017). In neurodegenerative disease and cancer, oxysterols interact with specific proteins and transcription factors, such as the liver X receptors (LXR) and oxysterol-binding protein and insulin-induced gene 1 (INSIG1) (Kloudova *et al*, 2017). Recently, it has been shown that 27OHC can inhibit prostate cancer growth through depletion of intracellular

1  Shandong Key Laboratory of Brain Function Remodeling, Department of Neurosurgery, Qilu Hospital of Shandong University and Institute of Brain and Brain-Inspired Science, Shandong University, Jinan, China
2  Department of Biomedicine, University of Bergen, Bergen, Norway
3  INSERM U1029, Institut Nationale de la Santé et de la Recherche Médicale, Pessac, France
4  University Bordeaux, Pessac, France
5  School of Medicine, Shandong University, Jinan, China
6  Department of Pathology, Haukeland University Hospital, Bergen, Norway
7  Department of Biomedicine, The Molecular Imaging Center, University of Bergen, Bergen, Norway
8  NorLux Neuro-Oncology Laboratory, Department of Oncology, Luxembourg Institute of Health, Luxembourg City, Luxembourg
   *Corresponding author. Tel: +47 55586042; E-mail: rolf.bjerkvig@uib.no
   **Corresponding author. Tel: +86 0537 82169428; E-mail: lixg@sdu.edu.cn
   ***Corresponding author. Tel: +86 0537 82169429; E-mail: jian.wang@uib.no
   †These authors contributed equally to this work
   ‡These authors contributed equally to this work as senior authors

cholesterol levels (Alfaqih *et al*, 2017). Yet, others have shown, in breast cancer cells, that 27OHC induces an epithelial-to-mesenchymal transition (EMT) leading to increased tumour growth (Torres *et al*, 2011; Wu *et al*, 2013). It has also been shown in a subtype of early-stage hepatocellular carcinoma that disrupted cholesterol homeostasis is associated with poor prognosis (Jiang *et al*, 2019). In GBM, the notion that excessive accumulation of cholesterol can fuel the growth of tumour cells has been supported by the following evidence. First, *de novo* cholesterol synthesis is suppressed in GBM cells compared with normal human astrocytes leading to exogenous cholesterol uptake through up-regulation of the low-density lipoprotein receptor (LDLR) (Villa *et al*, 2016). Second, loss of endogenous oxysterols in GBM leads to decreased cholesterol efflux (Villa *et al*, 2016). At present, the mechanisms causing dysregulation of cholesterol homeostasis in GBM are not clear, especially with regard to oxysterol loss.

In mammals, cholesterol homeostasis is tightly regulated by a complex protein network centred around two groups of transcription factors, sterol-regulatory binding proteins (SREBPs) and liver X receptors (LXRs), that are involved in cholesterol import, export, synthesis, metabolism and esterification (Ikonen, 2008). SREBPs mainly promote the transcription of genes, such as HMGCR and LDLR, that are involved in cholesterol synthesis and uptake when intracellular cholesterol levels decrease. LXRs respond to excessive intracellular cholesterol by inducing the expression of genes, such as *ABCA1* and *ABCG1*, that are responsible for cholesterol efflux and LDLR degradation through induction of the E3 ubiquitin ligase IDOL (Zelcer *et al*, 2009). Targeting these pathways has been shown to be an effective strategy for inhibiting growth in GBM animal models (An & Weiss, 2016).

Cancer cells have clearly developed mechanisms for an accumulation of excessive intracellular cholesterol in order to support their growth. Cholesterol metabolism might therefore represent a potential therapeutic target. However, the key factors contributing to the accumulation of cholesterol in GBMs remain unclear. Here, we used large-scale *in silico* analyses of whole-transcriptome databases to identify dysregulated genes in GBMs involved in cholesterol homeostasis. One of the most dramatically down-regulated genes was cholesterol 24-hydroxylase (CYP46A1), a brain-specific enzyme responsible for the elimination of cholesterol through conversion of cholesterol into 24(S)-hydroxycholesterol (24OHC) (Moutinho *et al*, 2016). *CYP46A1* expression emerged as a prognostic marker in GBM patients, and in functional studies, overexpression or pharmacological activation of the CYP46A1/24OHC axis suppressed GBM cell growth *in vitro* and *in vivo*. Our results show that changes in CYP46A1 are critical for the dysregulation of cholesterol homeostasis in GBM and that targeting CYP46A1/24OHC may provide a new opportunity for GBM therapy.

# Results

## Cholesterol level promotes GBM cell growth

First, we determined the difference in cholesterol regulation between GBM and normal brain. Gene set enrichment analysis (GSEA) based on the Rembrandt data revealed that GBM ($n = 217$) and normal brain tissues ($n = 28$) displayed a distinct enrichment

of cholesterol metabolic processes compared with normal brain (Appendix Fig S1A). Cholesterol synthesis, however, was positively associated with normal brain (adjusted $P < 0.05$, respectively; Appendix Fig S1A), which is consistent with previous reports indicating that GBM cells mainly depend on exogenous cholesterol synthesized by normal human astrocytes for growth (Villa *et al*, 2016). To confirm that GBM cells depend on exogenous cholesterol, two GBM cell lines (LN229 and U251) and GSCs from a primary GBM (GBM#P3) were cultured in media with FBS or in lipoprotein-deficient serum (LPDS). Growth curves generated revealed a significantly slower growth of GBM cells in LPDS after 5 days ($P < 0.01$). However, the addition of 5 μg/ml LDL (the lipoprotein with the highest cholesterol content) to the LPDS medium rescued GBM cell growth compared with LPDS medium (Appendix Fig S1B–D). These data confirm the importance of exogenous cholesterol for GBM cell growth as reported previously (Guo *et al*, 2011; Geng *et al*, 2016).

## CYP46A1 is a tumour suppressor candidate in GBM

To identify the most dysregulated cholesterol-related genes in GBM, we performed bioinformatic analysis on publicly available genomic datasets. First, we derived a gene signature of 176 genes involved in cholesterol biology based on Gene Ontologies (Alfaqih *et al*, 2017). Through differential analysis using the Rembrandt dataset, we uncovered a total of 13 differentially expressed genes between GBM ($n = 217$) and normal brain tissue ($n = 28$) (Fig 1A and B). The brain-specific cholesterol regulator, *CYP46A1*, emerged as the most differentially expressed gene with decreased expression in GBM ($\log_2$ fold change = 2.335, adjusted $P = 5.85E\text{-}25$). Differential analysis based on the Chinese Glioma Genome Atlas (CGGA) dataset also revealed *CYP46A1* as one of the most dysregulated transcripts ($\log_2$ fold change = 1.966, adjusted $P = 4.63E\text{-}09$) between GBM ($n = 128$) and normal brain ($n = 5$; Fig EV1A–C). Univariate Cox regression analysis in GBM patients was also used to screen the cholesterol-related genes based on prognostic values. After ranking by $P$-value, *CYP46A1* emerged among the top 3 genes (*APOBR*, *CELA3A* and *CYP46A1*) associated with GBM prognosis (Fig EV1D). By analysing the Cancer Genome Atlas (TCGA) pan-cancer data including 31 different cancer types, the expression of *CYP46A1* was found to be significantly increased in normal brain compared with GBM and LGG (Appendix Fig S2A). Loss of *CYP46A1* in GBM was further confirmed by analysing several public glioma datasets (over 1,500 samples were enrolled; $P < 0.01$, respectively; Appendix Fig S2B).

The intra-tumoral expression pattern of *CYP46A1* in GBMs was further determined using the IVY GBM RNA-seq data (http://glioblastoma.alleninstitute.org/). *CYP46A1* was highly expressed at the leading edge (which is mainly comprised of normal brain cells) compared with other tumour regions (Appendix Fig S2C). Single-cell RNA-seq data (Darmanis *et al*, 2017) further demonstrated that *CYP46A1* is mainly expressed in neurons, astrocytes and oligodendrocyte precursor cells (OPCs) and to a lesser extent in tumour cells (Appendix Fig S2D). CYP46A1 protein levels were also examined in different cell lines (Appendix Fig S2E). Normal human astrocytes (NHAs) displayed abundant CYP46A1 protein levels, while GBM cells (GBM#P3, GBM#05, GBM#BG7, LN229, U251 and LN18) showed much lower expression.

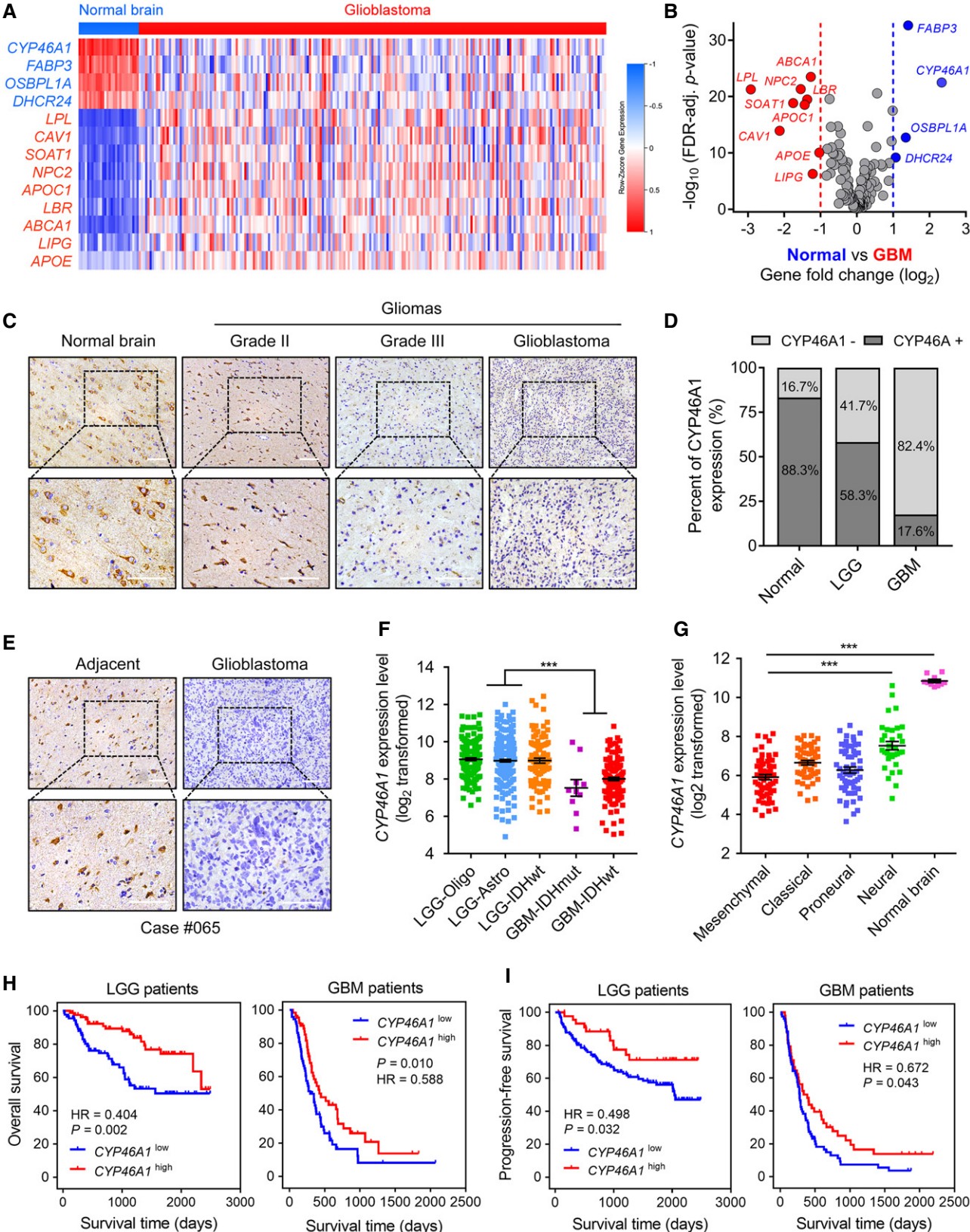

Figure 1.

◀

**Figure 1. Expression of *CYP46A1*, a gene involved in brain cholesterol homeostasis, is lost in GBM and associated with increasing tumour grade.**

A    Heatmap of the differentially expressed cholesterol-related genes between normal brain tissues (n = 28) and glioblastomas (n = 217) from the Rembrandt dataset. Gene expression values are z-transformed and coloured red for high expression and blue for low expression, as indicated in the scale bar.
B    Volcano plot showing the fold change (log2) in cholesterol-related gene levels based on GBM versus normal brain tissue. Data were obtained from the Rembrandt dataset.
C    Representative images of IHC staining for CYP46A1 protein in normal brain and different pathological grades of gliomas (n = 64). Scale bar = 30 μm.
D    Quantification of CYP46A1 IHC staining in normal brain (n = 6) and different pathological grades of gliomas (n = 58).
E    Representative images of CYP46A1 IHC staining in GBM and adjacent brain tissues from one specific case. Scale bar = 30 μm.
F    *CYP46A1* expression levels in tumours from the TCGA dataset using 2016 WHO classification. Data are shown as the mean ± the standard error of the mean (SEM; n = 667). ***$P$ < 0.0001. Statistical significance was determined by one-way ANOVA.
G    *CYP46A1* expression levels in different molecular subtypes from the Rembrandt GBM dataset. Shown are means and SEM (n = 245). ***$P$ < 0.0001. Statistical significance was determined by one-way ANOVA.
H–I  Kaplan–Meier analysis for patient OS and PFS based on high versus low expression of *CYP46A1* in LGG and GBM. Data were obtained from the CGGA dataset. $P$-values were obtained from the log-rank test.

To confirm that CYP46A1 expression is reduced in GBMs at the protein level, we performed IHC staining for CYP46A1 on an independent cohort of glioma (n = 58) and normal brain tissue samples (n = 6). CYP46A1 expression was found to be cytoplasmic. Expression was also consistently high in normal brain tissue (5/6; 88.3%) compared with LGG (14/24; 58.3%) and GBM (6/34; 17.6%), and decreased with increasing glioma grade (Fig 1C and D). IHC staining in one specific GBM sample demonstrated that CYP46A1 was positively expressed in adjacent brain tissue while absent in GBM (Fig 1E). IHC data from the Human Protein Atlas (n = 17) corroborated these results, confirming high expression of CYP46A1 in normal brain (Fig EV2A), which decreased with increasing tumour grade (Fig EV2B and C).

We subsequently assessed *CYP46A1* levels based on the 2016 WHO classification of gliomas, using the TCGA data. *CYP46A1* was higher in three LGG subtypes (LGG-Oligo, LGG-Astro and LGG-*IDH*wt) and lower in GBM-*IDH*mt and GBM-*IDH*wt (Fig 1F). Although there is a general down-regulation of CYP46A1 in GBM compared with normal brain, higher expression of *CYP46A1* was also observed in the Neural GBM molecular subtype (Fig 1G), which is associated with a more favourable prognosis, relative to the other subtypes based on the TCGA Verhaak-2010 molecular classification of GBM (Noushmehr *et al*, 2010).

To address why CYP46A1 is decreased in GBMs, we examined its promoter/enhancer regions using the combined chromatin immuno-precipitation sequencing (ChIP-seq) data from GEO and ENCODE databases. Here, H3K27ac peaks (marker of active promoters) within the promoter region of *CYP46A1* were lower in GBMs compared with normal brain tissue (Appendix Fig S3A). We also examined the active enhancer landscape of *CYP46A1* across three matched pairs of GSCs and differentiated glioma cells (DGCs). *CYP46A1* enhancers and mRNA levels tended to decrease in GSC versus DGC, as measured by ChIP-seq (H3K27ac and H3K4me3 peak levels) and mRNA data (Appendix Fig S3A–C). These results were also validated through ChIP-qPCR and Western blot analysis (Appendix Fig S3D and E). Taken together, abnormal histone modifications may partially explain reduced CYP46A1 expression in GBM.

## Decreased *CYP46A1* levels correlate with worse survival in glioma patients

To determine the clinical significance of CYP46A1, Kaplan–Meier analysis was performed using the CGGA dataset. GBM patients with

high *CYP46A1* mRNA levels (based on the median value) exhibited significantly better overall survival (OS) as well as progression-free survival (PFS) (Fig 1H and I). *CYP46A1* was also a prognostic indicator in LGG patients (Fig 1H and I). The prognostic value of *CYP46A1* was further validated in TCGA, Rembrandt and Phillips datasets (Appendix Fig S4A–C). *CYP46A1* was also validated as an independent prognostic indicator using univariate and multivariate Cox regression analysis of OS (HR = 0.390, 95% CI = 0.262 to 0.581, $P$ < 0.001; Appendix Table S1) in CGGA patients and showed a prognostic trend in multivariate analysis of TCGA patients. Collectively, these results emphasize the prognostic relevance of *CYP46A1* in GBM.

## Expression of CYP46A1 attenuates GBM growth

Next, we examined the role of CYP46A1 in GBM growth *in vitro*. Lentivirus was used to over-express CYP46A1 in LN229, LN18 and GBM#P3, as validated by qRT–PCR ($P$ < 0.001; Fig 2A) or Western blot (Fig 2B and Appendix Fig S5A). Cell growth was significantly inhibited in lenti-CYP46A1 compared with lenti-Ctrl cells ($P$ < 0.01; Fig 2C). Ectopic expression of CYP46A1 also attenuated colony formation ($P$ < 0.05; Fig 2D). Furthermore, increased expression of CYP46A1 in GSCs (GBM#P3) inhibited tumorsphere formation, a critical glioma stem-like property ($P$ < 0.05; Appendix Fig S5B and C).

LN229 and GBM#P3-lenti-CYP46A1 or GBM#P3-lenti-control cells were injected intracranially into mice to evaluate tumour growth *in vivo*. For both cell types, CYP46A1 overexpression inhibited tumour growth, as observed in haematoxylin and eosin (HE) staining (Fig 2E), and prolonged overall survival in mice ($P$ < 0.05; Fig 2F). Overexpression of CYP46A1 also led to reduced levels of the proliferation marker PCNA but increased expression of the apoptotic marker cleaved caspase-3 (Fig 2G, Appendix Fig S5D and E). These data confirm a tumour-suppressive role for CYP46A1 in human GBM.

## 24OHC inhibits GBM growth and GSC maintenance

The *CYP46A1* gene encodes a cytochrome P450 oxidase, sterol 24-hydroxylase, the main function of which is to convert cholesterol into 24OHC (Fig 3A). We therefore characterized metabolic changes in 24OHC in human GBM. 24OHC levels were decreased in primary GBMs (n = 3) compared with normal brain tissue samples

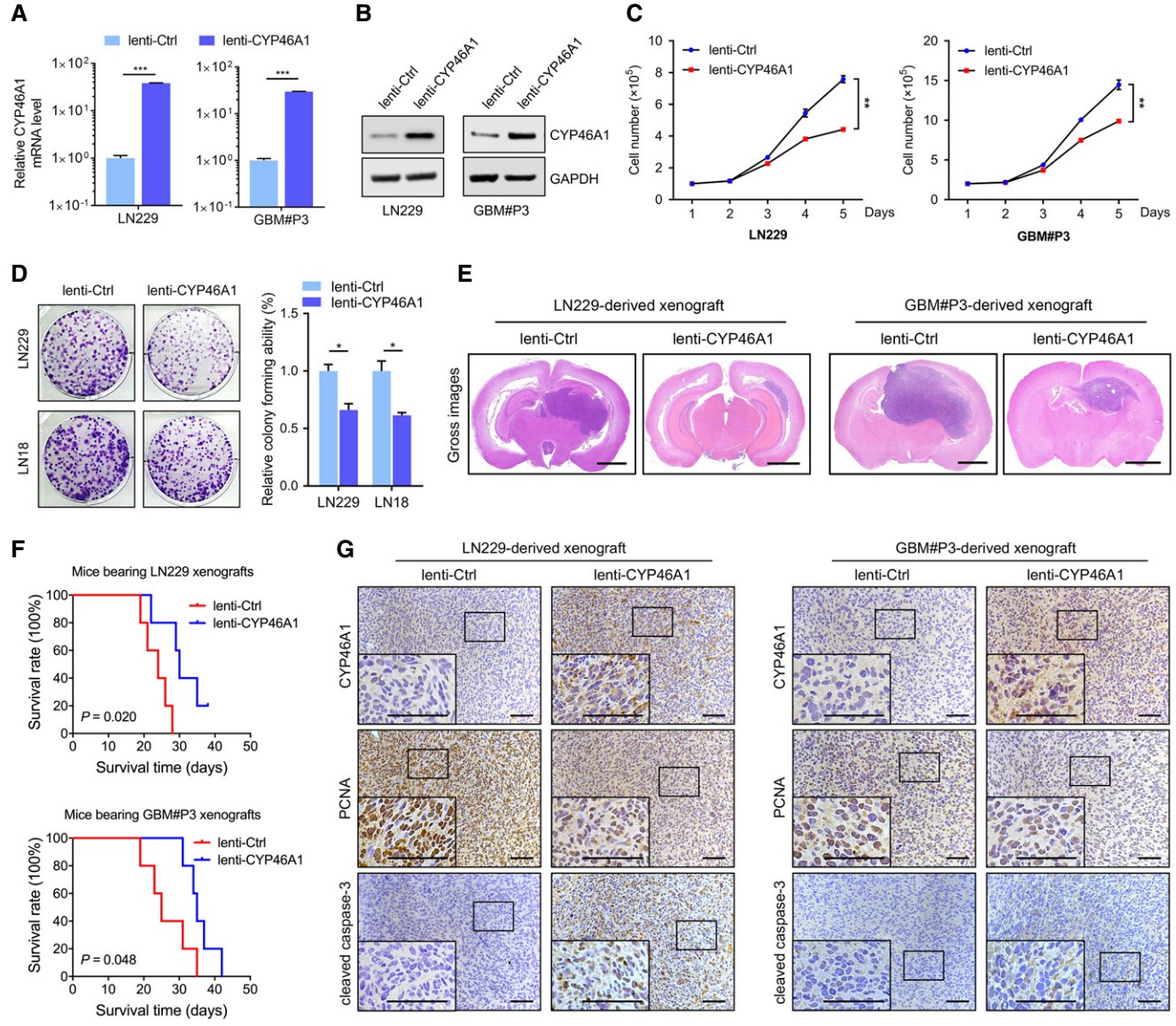

**Figure 2. Restoration of CYP46A1 attenuates the growth of GBM cells *in vitro* and in intracranial GBM xenografts.**

A   qRT–PCR analysis of *CYP46A1* mRNA levels in LN229 and GBM#P3 cells transfected with lentivirus expressing CYP46A1 (lenti-CYP46A1) or control sequence (lenti-Ctrl). *GAPDH* was used as an internal control. Shown are means and SEM (*n* = 3). LN229: ***$P < 0.0001$; GBM#P3: ***$P < 0.0001$. Statistical significance was determined by two-sided Student's *t*-test.
B   Western blot analysis to confirm CYP46A1 overexpression in GBM cells.
C   Growth curves for GBM cells *in vitro* infected with lenti-Ctrl or lenti-CYP46A1 derived from trypan blue staining. Shown are means and SEM (*n* = 3). LN229: **$P = 0.003$; GBM#P3: **$P = 0.002$. Statistical significance was determined by two-sided Student's *t*-test.
D   Colony-forming assay for GBM cells infected with lenti-Ctrl or lenti-CYP46A1. Shown are means and SEM (*n* = 3). LN229: *$P = 0.013$; LN18: *$P = 0.012$. Statistical significance was determined by two-sided Student's *t*-test.
E   Representative H&E staining of orthotopic tumours derived from GBM cells infected with lenti-Ctrl or lenti-CYP46A1. Scale bar = 2 mm.
F   Kaplan–Meier survival curve of tumour-bearing mice injected with GBM cells infected with lenti-Ctrl or lenti-CYP46A1 (*n* = 5 per group). A log-rank test was used to assess statistical significance.
G   IHC for CYP46A1, PCNA and cleaved caspase-3 protein levels in the intracranial tumours. Scale bar = 100 μm.

($P < 0.05$; Appendix Fig S6A). These results were consistent with decreased *CYP46A1* expression in GBM. Ectopic expression of CYP46A1 in GBM cells *in vitro*, however, led to a dramatic increase in 24OHC concentration in culture media (Fig 3B) as well as in GBM cell pellets ($P < 0.01$; Appendix Fig S6B). Exogenous 24OHC

inhibited LN229 growth in a dose- and time-dependent manner (Appendix Fig S6C and D). IC$_{50}$s for 24OHC were 28.86 μM in LN229 and 33.81 μM in GBM#P3 cells (Fig 3C).

To identify potential mechanisms for 24OHC inhibition of GBM proliferation, cell cycle parameters and apoptosis were determined

using flow cytometry (analysis of Annexin V-FITC and PI staining). Treatment of LN229 and GBM#P3 with 24OHC increased apoptosis in a dose-dependent manner (Fig 3D and E). 24OHC also elevated caspase-3/7 activity in LN229 (Appendix Fig S6E and F) and dramatically inhibited colony formation in LN229 and LN18 compared with controls ($P < 0.001$, Fig 3F). Furthermore, Western blot analysis showed an increase in cleaved PARP and cleaved caspase-3, and a decrease in PCNA in a time-dependent manner (Fig 3G and Appendix Fig S6G). Cell cycle analysis based on PI staining indicated only a slight increase in G2/M phase cells after 24OHC treatment (Appendix Fig S6H). Finally, treatment of GSCs, GBM#P3, GBM#BG7 and GBM#BG5, with 24OHC, led to a reduced tumorsphere formation (Fig 3H and I). In contrast, 24OHC was less toxic to normal human astrocytes (NHAs) and rat brain organoids (Fig EV3A–C). In summary, these data show that 24OHC specifically inhibits GBM growth.

### 24OHC suppresses the growth of GBM cells by reducing cholesterol accumulation

It has been shown that neurons and astrocytes both produce oxysterols, including 24OHC, as a result of brain cholesterol metabolism. These molecules act as endogenous ligands for the liver X receptors (LXRs) to decrease excess cellular cholesterol levels (Repa et al, 2000; Venkateswaran et al, 2000; Zelcer et al, 2009; Chen et al, 2013). In GBM cells in vitro, 24OHC treatment resulted in a significant decrease in total intracellular cholesterol levels ($P < 0.01$, Fig 4A) and filipin staining which reflects free cholesterol levels (Fig 4B). These results therefore show that 24OHC inhibits GBM cell growth by depleting intracellular cholesterol. To further validate 24OHC induced growth inhibition, rescue experiments with cholesterol were performed in LN229 cells. The addition of exogenous cholesterol significantly increased the growth of 24OHC treated cells ($P < 0.0001$; Fig 4C). Cholesterol also attenuated apoptosis caused by 24OHC in both LN229 and GBM#P3 (Fig 4D) and decreased cleaved PARP (Fig 4E). Cholesterol also compensated for the reduction in GSC sphere formation caused by 24OHC in both GBM#P3 and GBM#BG7 cells ($P < 0.05$; Fig 4F). Finally, exposure to methyl-β-cyclodextrin (MβCD), which depletes cellular cholesterol, also inhibited GBM cell survival ($P < 0.001$ at 10 mM; Fig 4G). In conclusion, these results demonstrated that 24OHC suppresses GBM growth by depleting cellular cholesterol.

### 24OHC inhibits GBM growth through regulation of LXR and SREBP1 activity

To determine the mechanisms underlying 24OHC inhibition of GBM growth, we performed RNA sequencing (RNA-seq) of GBM#P3 cells treated with or without 24OHC (heatmap Fig 5A). GO analysis indicated that differentially expressed genes were closely related to biological processes involved in negative regulation of lipid storage and positive regulation of cholesterol efflux, cholesterol homeostasis and cell apoptosis. In addition, inflammatory signalling pathways including IL-10 and NF-κB pathways were regulated by 24OHC (Fig 5B). KEGG pathway analysis also indicated that these genes were strongly associated with the LDLR pathway, the FXR (farnesoid X receptor) and LXR regulation of cholesterol metabolism (Fig 5C). Moreover, GSEA showed that 24OHC was associated with positive regulation of cholesterol efflux, but negatively associated with gene signatures linked to cholesterol homeostasis, SREBP targets and stem cell proliferation (Fig 5D). A volcano plot of the RNA-seq results revealed significant changes in several target genes of two cholesterol homeostasis transcription factors, LXR and SREBPs. Interestingly, GSC stemness markers, including SOX2 and SOX4, decreased after exposure to 24OHC (Fig 5E), which was consistent with a decreased tumorsphere formation ability. RNA-seq results for these genes were further confirmed by qRT–PCR (Fig 5F) and immunofluorescence staining of SOX2 in GBM#P3 tumorspheres treated with 24OHC (Appendix Fig S7A).

24OHC is an endogenous LXR agonist (Chen et al, 2013). We found that 24OHC triggered GBM cell death in the presence of LXR-623 ($P < 0.05$; Fig 5G), which is an effective synthetic LXR agonist (Villa et al, 2016). These results indicate that 24OHC might kill GBM cells not only through activation of LXR, but also through other mechanisms. Based on the RNA-seq data, 24OHC was also found to suppress SREBP signalling (Fig 5D and E). Analysis of the genomic datasets consistently showed that SREBP1 was the predominant SREBP isotype expressed in GBMs predicting poor overall survival (Appendix Fig S7B). This is consistent with prior reports showing that SREBP1, but not SREBP2, contributes to GBM progression and tumorigenesis (Geng et al, 2016). We thus reasoned that 24OHC influenced SREBP1 transcriptional activity in GBM. On Western blots, proteins levels were correspondingly altered; 24OHC caused a decrease in protein levels of nuclear SREBP1 (N-SREBP1), precursor SREBP1 (P-SREBP1) and LDLR as well as an induction of

---

**Figure 3. CYP46A1 inhibits the growth of GBM cells through the production of 24OHC.**

A   Schematic model of cholesterol conversion into 24OHC catalysed by CYP46A1.

B   24OHC levels in spent media from LN229 (left) and GBM#P3 (right) cells transduced with lenti-Ctrl or lenti-CYP46A1 measured using targeted LC-MS/MS. Data are shown as the mean ± SEM (n = 3). Statistical significance was determined by two-sided Student's t-test (see Materials and Methods).

C   IC50 curves for 24OHC in LN229 and GBM#P3 cells using the CCK-8 assay.

D, E   Flow cytometry to detect Annexin V-FITC and PI staining to determine the percentage of LN229 and GBM#P3 cells undergoing apoptosis after exposure to 0–20 μM 24OHC for 72 h. Data are shown as the mean ± SEM (n = 3). LN229: ***P = 0.0005, ***P < 0.0001, ***P < 0.0001; GBM#P3: *P = 0.035, ***P = 0.0006, ***P < 0.0001. Statistical significance was determined by one-way ANOVA.

F   Colony-forming ability of GBM cell lines treated with 24OHC (0–20 μM) for 14 days. Data are shown as the mean ± SEM (n = 3). ***P < 0.0001. Statistical significance was determined by one-way ANOVA.

G   Western blot analysis of the apoptosis marker c-PARP, c-caspase-3 and PCNA in lysates (20 μg) from GBM cells treated with 24OHC (0–20 μM) for 72 h.

H–I   Tumorsphere formation assays for GSCs treated with different concentrations of 24OHC (0–20 μM). Scale bar = 100 μm. Graphic representation of the quantification of tumorsphere formation. Data are shown as the mean ± SEM (n = 3). GBM#P3: *P = 0.0192, ***P = 0.0006, ***P = 0.0002; GBM#BG7: **P = 0.0017, ***P < 0.0001, ***P < 0.0001; GBM#BG5: **P = 0.0027, ***P = 0.0001, ***P < 0.0001. Statistical significance was determined by one-way ANOVA.

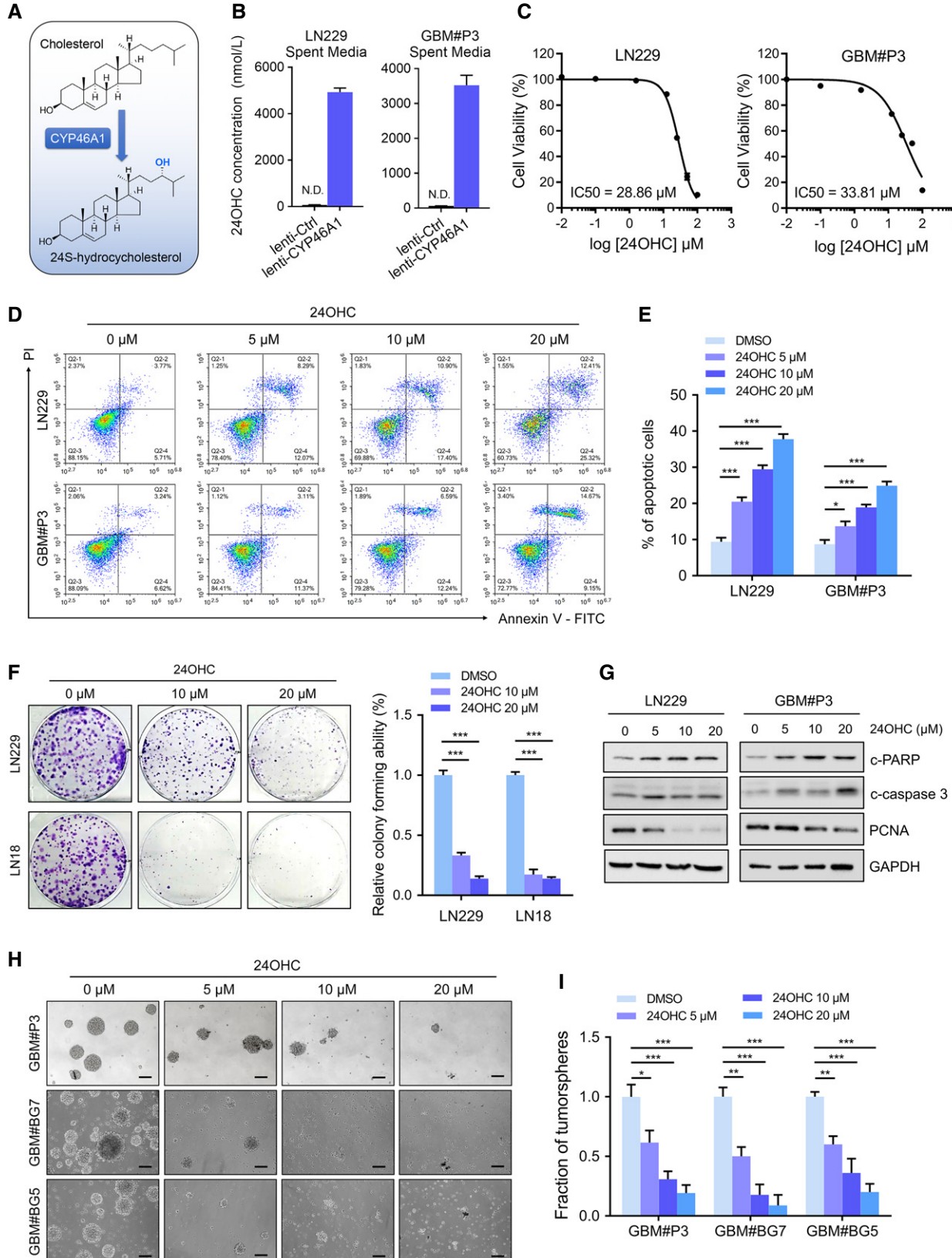

**Figure 3.**

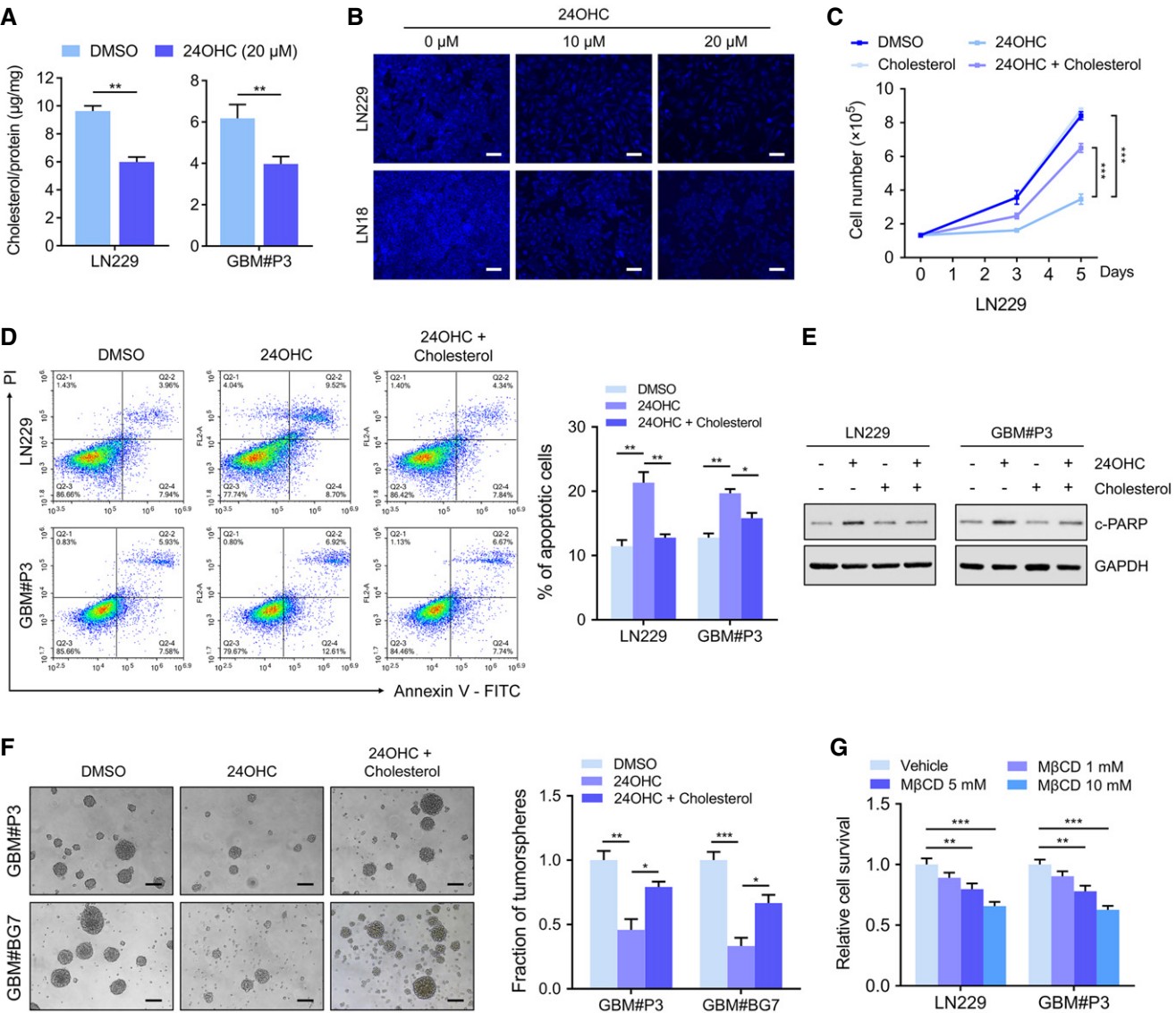

**Figure 4. 24OHC suppresses the growth of GBM cells by reducing cholesterol content.**

A   Intracellular levels of cholesterol in GBM cells treated with 20 μM of 24OHC or DMSO for 72 h quantified using the Invitrogen™ Amplex™ Red Cholesterol Assay Kit and normalized to total protein. Shown are means and SEM (n = 3). LN229: **P = 0.002; GBM#P3: **P = 0.011. Statistical significance was determined by two-sided Student's t-test.

B   Representative images of filipin staining in GBM cells treated with different concentrations of 24OHC (0–20 μM) for 72 h. Scale bar = 30 μm.

C   Growth curves for LN229 cells treated with DMSO, 0.5 μg/ml cholesterol or 10 μM 24OHC in the presence or absence of 0.5 μg/ml cholesterol using trypan blue staining. Data are shown as the mean ± SEM (n = 3). ***P < 0.0001. Statistical significance was determined by one-way ANOVA.

D   Flow cytometry to detect Annexin V-FITC and PI staining to determine the percentage of GBM cells undergoing apoptosis after exposure to DMSO or 10 μM 24OHC in the presence or absence of 0.5 μg/ml cholesterol for 72 h. Data are shown as the mean ± SEM (n = 3). LN229: **P = 0.0021, **P = 0.0046; GBM#P3: **P = 0.0012, *P = 0.0193. Statistical significance was determined by one-way ANOVA.

E   Western blot analysis of c-PARP in GBM cells treated with DMSO or 10 μM 24OHC in the presence or absence of 0.5 μg/ml cholesterol for 72 h.

F   Tumorsphere formation assays for GBM#P3 and GBM#BG7 treated with DMSO or 10 μM 24OHC in the presence or absence of 0.5 μg/ml cholesterol. Scale bar = 100 μm. Data are shown as the mean ± SEM (n = 3). GBM#P3: **P = 0.0032, *P = 0.031; GBM#BG7: ***P = 0.0008, *P = 0.0242. Statistical significance was determined by one-way ANOVA.

G   CCK-8 assay to determine relative cell survival in LN229 or GBM#P3 cells after treatment with vehicle control or different concentrations of MβCD for 72 h. Data are shown as the mean ± SEM (n = 3). LN229: **P = 0.0051, ***P = 0.0002; GBM#P3: **P = 0.0018, ***P < 0.0001. Statistical significance was determined by one-way ANOVA.

ABCA1 (Fig 5H and Appendix Fig S7C). More importantly, overexpression of SREBP1 partially rescued 24OHC-induced growth inhibition of LN229 cells (P < 0.01, Fig 5I). Furthermore, a specific SREBP antagonist, fatostatin, suppressed the growth of LN229 and LN18 cells *in vitro* (P < 0.001, Fig 5J) and GBM#P3 tumour growth *in vivo* (n = 3 per group; 15 mg/kg/day with intraperitoneal injection;

Appendix Fig S10D). Finally, filipin staining confirmed that cholesterol levels in GBM cells decreased under fatostatin treatment (Fig 5K).

These results show that inhibition of cholesterol leading to reduced cell growth by 24OHC is partly mediated by up-regulation of LXR target genes and down-regulation of SREBP1 activity.

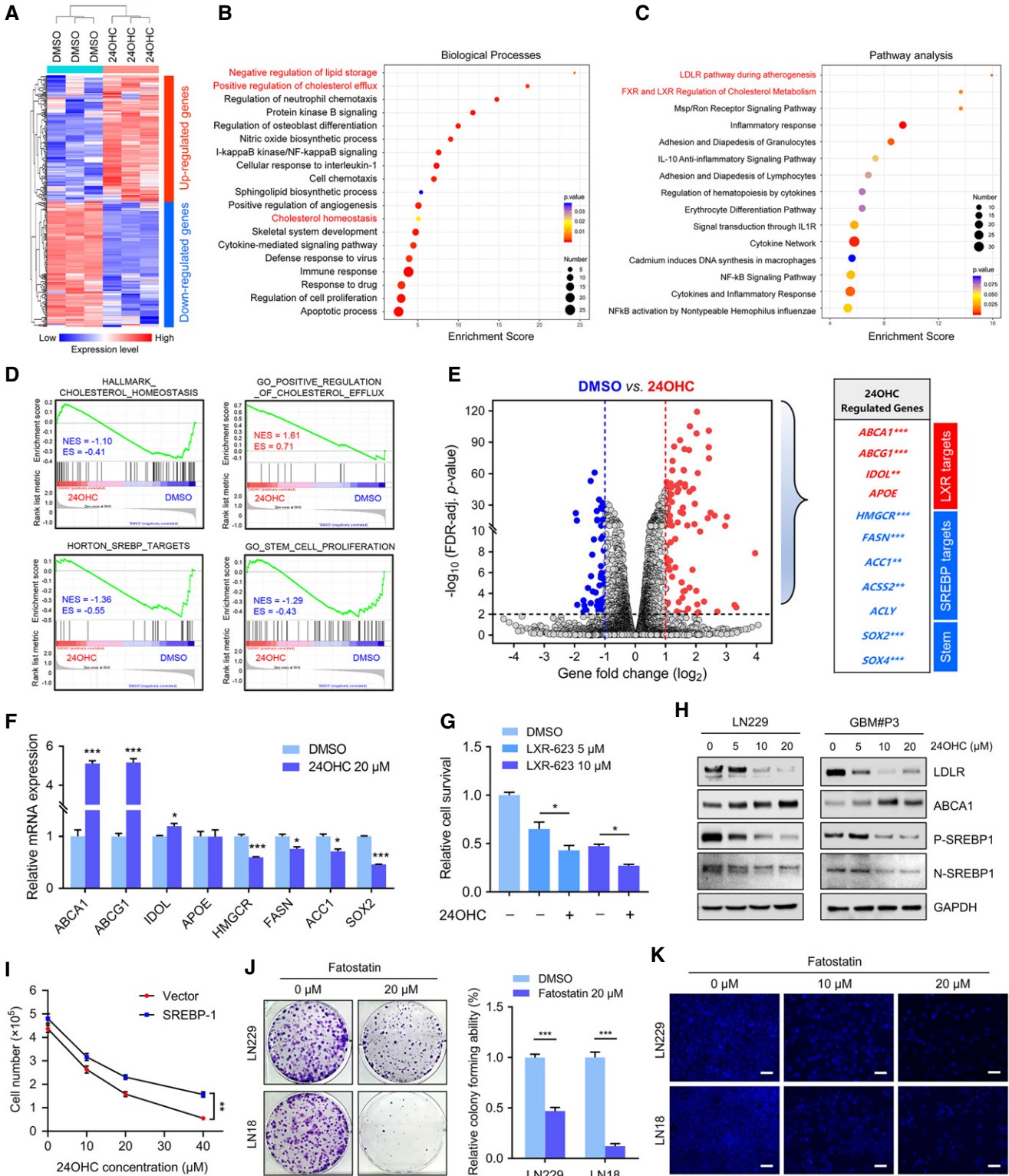

**Figure 5.**

Figure 5. The downstream mechanisms of 24OHC in GBM progression.

A  Heatmap of differentially expressed transcripts in RNA-seq data from GBM#P3 cells treated with DMSO or 20 μM 24OHC for 72 h. Gene expression data were mean z-transformed for display and coloured red for high expression and blue for low expression.

B, C  GO and KEGG enrichment analysis of differentially expressed genes based on RNA-seq data with adjusted P-value.

D  GSEA plot showing normalized enrichment scores (NESs) for cholesterol homeostasis, positive regulation of cholesterol efflux, SREBP targets and stem cell proliferation signatures using RNA-seq data from 24OHC and DMSO-treated cells.

E  Volcano plot highlighting up-regulated (red) and down-regulated (blue) genes based on RNA-seq data from GBM#P3 cells after 24OHC treatment. LXR targets, SREBP targets and stemness transcription factors are specifically included. **adjusted P < 0.01, ***adjusted P < 0.001.

F  qRT-PCR analysis of 24OHC-regulated genes in GBM#P3 cells after treatment with 20 μM 24OHC for 72 h. GAPDH was used as an internal control. Data are shown as the mean ± SEM (n = 3). ***P < 0.0001, ***P < 0.0001, *P = 0.0192, ***P = 0.0004, *P = 0.0118, *P = 0.0101, ***P < 0.0001. Statistical significance was determined by two-sided Student's t-test.

G  Cell viability assay performed on GBM#P3 cells after 3 days of treatment with DMSO or 5–10 μM LXR-623 in the presence or absence of 10 μM 24OHC. Data are shown as the mean ± SEM (n = 3). *P = 0.031, *P = 0.050. Statistical significance was determined by one-way ANOVA.

H  Western blot analysis of cholesterol metabolism-related proteins in LN229 or GBM#P3 after treatment with 24OHC (0–20 μM).

I  Growth curves for LN229 cells transfected with control vector or SREBP1 plasmid and treated with 24OHC (0-40 μM) as assessed by a trypan blue test. Data are shown as the mean ± SEM (n = 3). **P = 0.0039. Statistical significance was determined by two-sided Student's t-test.

J  Colony-forming ability of GBM cell lines treated with DMSO or 20 μM fatostatin for 14 days. Data are shown as the mean ± SEM (n = 3). ***P < 0.0001. Statistical significance was determined by two-sided Student's t-test.

K  Representative images of filipin staining in LN229 (up) or LN18 (down) cells treated with different concentration of fatostatin (0–20 μM). Scale bar = 30 μm.

## Efavirenz inhibits GBM growth and prolongs survival of intracranial tumour-bearing mice

Based on the findings that CYP46A1 inhibits GBM growth by converting cholesterol to 24OHC, we investigated the effect of efavirenz (EFV) (Fig 6A), which is an anti-HIV medication known to activate CYP46A1 activity through binding to the enzyme's allosteric site (Mast et al, 2014, 2017a; Anderson et al, 2016). Bioinformatic chemical–protein interaction network analysis (STITCH) indicated a strong association between EFV and CYP46A1 protein, as well as PCNA (Appendix Fig S8A). EFV treatment reduced intracellular levels of cholesterol in GBM cells (Fig 6B and C). The $IC_{50}$ values for EFV were 16.81 μM in LN229 and 24.58 μM in GBM#P3 (Fig 6D). Also, increased levels of 24OHC were observed in culture media from EFV-treated cells (P < 0.05, Fig 6E).

Functional studies showed that EFV potently reduced LN229 and GBM#P3 tumour growth in a time- and dose-dependent manner (Appendix Fig S8B–D). Colony formation assays confirmed the toxicity of EFV in LN229 and LN18 (P < 0.01; Fig 6F). Flow cytometry analysis demonstrated that EFV induced apoptosis in GBM cells (P < 0.001, Fig 6G), while sparing normal brain organoids and NHAs (Appendix Fig S9A–C). In GSCs (GBM#P3, GBM#BG7 and GBM#BG5), EFV significantly suppressed tumorsphere formation (Fig 6H). At the molecular level, EFV led to increased levels of the LXR target ABCA1 and the apoptosis marker cleaved PARP, but to decreased levels of SREBP1 and LDLR, as well as PCNA and SOX2 (Fig 6I). Finally, exogenous cholesterol treatment (Appendix Fig S9D) or CYP46A1 knockdown (Appendix Fig S9E and F) partially restored EFV-induced growth inhibition of GBM cells. These results suggest that the anti-cancer effect of EFV is mediated through regulation of cholesterol levels.

The penetration of the blood–brain barrier (BBB) by EFV has been well-documented in several studies. Median levels of EFV concentration in the cerebrospinal fluid (CSF) were reported to be 11.1–30 μg/l (Tashima et al, 1999; Best et al, 2011; Tovar-y-Romo et al, 2012; Yilmaz et al, 2012; Calcagno et al, 2015; Winston et al, 2015). EFV (0.09 mg/kg/day) by oral gavage treatment in C57BL/6J mice, which is 6,667-fold lower than the dose given to HIV patients (600 mg/kg/day), was recently reported to be sufficient to activate

CYP46A1 and to generate a 42% increase in 24OHC in mouse brain tissue (Mast et al, 2014). Therefore, we investigated the therapeutic impact of EFV in mice bearing LN229 and GBM#P3 orthotopic xenografts (Fig 6J). Oral administration of EFV (0.1 mg/kg/day) was given to mice bearing tumours (n = 5 per group). Overall survival of mice was prolonged (median survival: 29 versus 35 days for GBM#P3; 27 versus 44 days for LN229; Fig 6K). For GBM#P3, bioluminescence imaging revealed a significant inhibition of tumour growth at day 21 (P < 0.01, Fig 6L and M). A trend for increasing body weight was also observed in EFV-treated animals (Fig 6N). No cytotoxic effect of EFV was observed during treatment. IHC staining of brain sections from tumour-bearing mice also demonstrated that EFV suppressed the expression of PCNA and induced protein levels of cleaved caspase-3 (Appendix Fig S10A–D). In conclusion, these findings demonstrate that EFV represents a promising drug repurposing strategy for GBM treatment.

## Discussion

Despite varying genetic backgrounds, several solid tumours including GBM rely on lipid metabolism to fuel tumour growth (Cheng et al, 2018). In the present study, GSEA of genomic data revealed an accumulation of cholesterol in GBMs compared with normal brain tissue. Consistent with previous reports (Guo et al, 2011; Geng et al, 2016), we showed that GBM cells require extracellular cholesterol to fuel tumour growth, indicating that a metabolic reprogramming and cholesterol homeostasis dysfunction occurs in GBM. Using a bioinformatic approach based on publicly available genomic data, we identified the most dysregulated cholesterol-related gene, CYP46A1, and investigated its role in GBM. The expression of CYP46A1 was negatively correlated with WHO grade and malignant clinicopathological features of gliomas. Moreover, through large-scale survival analysis, we showed that low CYP46A1 expression in glioma tissues is associated with poor prognosis and can therefore serve as an independent prognostic indicator.

Human CYP46A1, which is located on chromosome 14q32, contains 15 exons and encodes for cholesterol 24-hydroxylase (Lund et al, 1999). CYP46A1 is specifically expressed in the brain and is

responsible for the conversion of cholesterol into 24 (S)-hydroxy-cholesterol (24OHC), which crosses the BBB into the systemic circulation for metabolism (Lutjohann *et al*, 1996; Bjorkhem *et al*, 1997, 1998). The enzyme is responsible for at least 40–50% of brain cholesterol turnover (Lutjohann *et al*, 1996; Xie *et al*, 2003). Dysregulation of CYP46A1 expression has been shown to occur in several neurodegenerative diseases, including Parkinson's, Alzheimer's and Huntington diseases (Bjorkhem *et al*, 2013; Leoni *et al*, 2013; Soncini *et al*, 2016; Swan *et al*, 2016). However, its role in cancer progression is less understood (Soncini *et al*, 2016). In GBM, abnormal histone modifications such as acetylation and methylation are closely related to tumorigenesis and drug resistance (Romani *et al*,

2018). In the present study, we found lower levels of histone modification of H3K4me3 and H3K27ac sites at the promoter region of *CYP46A1*, which may partially account for the decreased expression of *CYP46A1* in GBM relative to normal brain. In addition, it has been reported that both the demethylating agent 5′-aza-2′-deoxycytidine (DAC) and the histone deacetylase inhibitor (HDACi) trichostatin A (TSA) can induce *CYP46A1* mRNA levels (Milagre *et al*, 2010; Nunes *et al*, 2010). Further studies of the upstream regulation of *CYP46A1* are therefore warranted.

In functional assays, we observed that overexpression of CYP46A1 repressed GBM proliferation both *in vitro* and *in vivo*. CYP46A1 accelerates the conversion of cholesterol into 24OHC

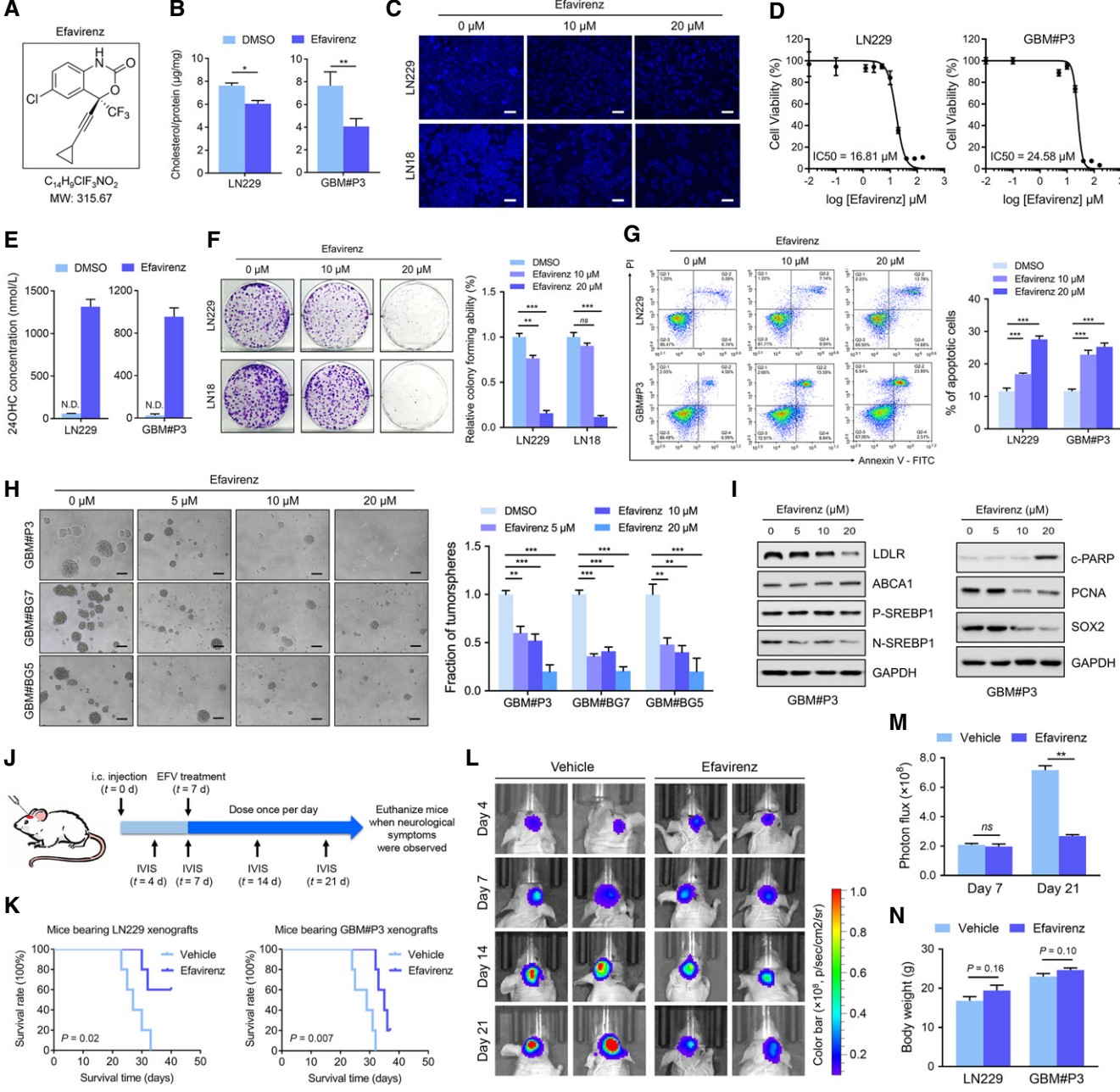

**Figure 6.**

Figure 6. Efavirenz, an activator of CYP46A1, attenuates the growth of GBM cells and prolongs survival of mice bearing intracranial tumours.

A    The molecular structure of EFV.
B    Intracellular concentrations of cholesterol in GBM cells after treatment with 20 μM EFV or DMSO and normalized to total protein. Data are shown as the mean ± SEM (n = 3). LN229: *P = 0.013; GBM#P3: **P = 0.0023. Statistical significance was determined by two-sided Student's t-test.
C    Filipin staining of GBM cells after treatment with EFV (0–20 μM). Scale bar = 30 μm.
D    IC50 curves for EFV in GBM cells determined using OD readings (450 nm) from the CCK-8 assay. Data are shown as mean ± SEM (n = 3).
E    24OHC levels in spent media from LN229 (left) or GBM#P3 (right) cells treated with DMSO or 20 μM EFV for 72 h quantified using targeted LC-MS/MS. Data are shown as the mean ± SEM (n = 3).
F    Images and quantification of colonies formed by GBM cell lines after treatment with different concentrations of EFV (0–20 μM). Data are shown as the mean ± SEM (n = 3). LN229: **P = 0.0056, ***P < 0.0001; LN18: ***P < 0.0001. Statistical significance was determined by one-way ANOVA.
G    Flow cytometry to detect Annexin V-FITC and PI staining to determine the percentage of GBM cells undergoing apoptosis after treatment with EFV (0–20 μM) for 72 h. Data are shown as the mean ± SEM (n = 3). LN229: ***P = 0.0006, ***P < 0.0001; GBM#P3: ***P < 0.0001. Statistical significance was determined by one-way ANOVA.
H    Tumorsphere formation assays for GSCs treated with different concentrations of EFV (0–20 μM). Scale bar = 100 μm. Graphic representation of the quantification of tumorsphere formation (right). Data are shown as the mean ± SEM (n = 3). GBM#P3: **P = 0.0011, ***P = 0.0003, *** P < 0.0001; GBM#BG7: ***P < 0.0001; GBM#BG5: **P = 0.0037, **P = 0.0015, ***P = 0.0002. Statistical significance was determined by one-way ANOVA.
I    Western blot analysis of cholesterol homeostasis-related proteins (LDLR, ABCA1, P-SREBP1 and N-SREBP1), c-PARP, PCNA and SOX2 in GBM cells treated with EFV (0–20 μM) for 72 h.
J    Schematic diagram of the schedule for implantation and drug treatment in the GBM xenograft model. Seven days after implantation of tumour cells, mice were treated with EFV by gavage (0.1 mg/kg/day). Bioluminescent imaging (BLI) using IVIS was performed at days 4, 7, 14 and 21.
K    The survival curves of tumour-bearing mice implanted with LN229 or GBM#P3 cells after EFV or PBS treatment (n = 5 per group). A log-rank test was used to assess statistical significance.
L, M  Bioluminescent images and the corresponding quantification of tumours in mice implanted with GBM#P3 cells at days 7 (n = 5 per group, P = 0.58) and 21 (n = 5 per group, P = 0.005). Data are shown as mean ± SEM. Statistical significance was determined by two-sided Student's t-test.
N    Body weight of tumour-bearing mice after 3 weeks of EFV or PBS treatment. Data are shown as the mean ± SEM (n = 5 per group). Statistical significance was determined by two-sided Student's t-test.

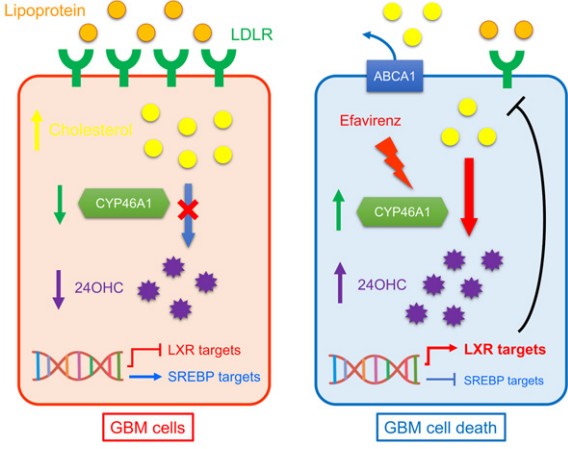

Figure 7.   Schematic representation of CYP46A1 contribution in GBM progression.

(Left panel) GBM cells have an excessive cholesterol accumulation caused by CYP46A1 down-regulation, which leads to a reduced 24OHC production that in turn cause a suppression of LXR signalling and an induction of SREBP targets. (Right panel) Restoration of CYP46A1 activity by EFV impairs GBM tumorigenic properties by increasing 24OHC levels, which in turns decreases intracellular cholesterol levels. Mechanistically, EFV treatment leads to the up-regulation of LXR target genes including *ABCA1*, and down-regulation of SREBP1 activity and LDLR expression.

(Lehmann *et al*, 1997; Janowski *et al*, 1999), such as the ATP-binding cassette transporter A1 (ABCA1) in both neurons and glia (Fukumoto *et al*, 2002) and apoE in astrocytes (Liang *et al*, 2004). RNA-seq revealed that 24OHC caused an up-regulation of LXR targets, such as *ABCA1*, *ABCG1* and *IDOL* (the E3 ligase of LDLR), and down-regulation of several SREBP target genes. In summary, these data show that 24OHC acts, at least partially, by regulating the activity of two essential transcription factors, LXR and SREBP, involved in cholesterol homeostasis. Although 24OHC likely modulates cell death also through other pathways (Noguchi *et al*, 2015), our results show that inhibition of CYP46A1/24OHC is a critical underlying regulator of cholesterol homeostasis in GBM. Thus, restoration of the CYP46A1/24OHC axis is a promising GBM treatment strategy.

Structure analysis of CYP46A1 suggested that distinct compounds can bind the active site of the enzyme (Mast *et al*, 2008). EFV, a BBB penetrating drug that has been used for the treatment of HIV, was found to bind to the CYP46A1 allosteric site. The drug alters the shape of the active site of cholesterol-free CYP46A1, making cholesterol-binding tighter and enzyme catalysis more efficient (Mast *et al*, 2014). Supporting its CNS activity, oral administration has been shown to improve long-term spatial memory, decrease amyloid-β content and also prolong survival of Alzheimer's mice (Mast *et al*, 2014, 2017b; Petrov *et al*, 2019). More recently, EFV has been shown to reduce phosphorylated Tau (pTau) accumulation in a human iPSC-derived AD model (van der Kant *et al*, 2019). EFV has been reported to exert anti-tumour effects in some cancer types, including renal clear-cell carcinoma, prostate cancer and ovarian cancer (Landriscina *et al*, 2008; Houede *et al*, 2014; Perna *et al*, 2017). However, the efficacy of EFV in the treatment of GBM remains unknown. In the current study, we observed that EFV (20 uM) significantly suppressed GBM growth and induced tumour cell death. At the molecular level, EFV stimulated the activity of CYP46A1, which resulted in elevated levels of 24OHC leading to a

through its enzyme activity. Further experiments indicated that 24OHC inhibits the growth of GBM through depletion of intracellular cholesterol. Therefore, the CYP46A1/24OHC axis suppresses proliferation of GBM by disturbing cholesterol homeostasis. Previous studies have identified 24OHC as an activator of LXR and inducer of several genes involved in cellular cholesterol efflux

decreased accumulation of cholesterol, accompanied by increased levels of the LXR target ABCA1, and decreased levels of SREBP1 and LDLR. Notably, exogenous cholesterol treatment or CYP46A1 knockdown partially restored growth inhibition of GBM cells induced by EFV, suggesting that the anti-cancer effect of EFV is mediated at least partially, by regulating cholesterol levels. EFV has been frequently associated with neuropsychiatric adverse events (NPAE) when given to HIV patients at high doses (Apostolova *et al*, 2017). However, major NPAE were not observed in mice at lower doses of EFV, indicating an appropriate therapeutic window (Mast *et al*, 2017b). Finally, EFV showed promising anti-glioma effects both *in vitro* and *in vivo*. A schematic representation of our results is illustrated in Fig 7.

We show that accumulation of cholesterol and dysregulated cholesterol homeostasis in GBM is mediated by loss of CYP46A1 and that CYP46A1 represents a viable therapeutic target in GBMs. An activator of CYP46A1 inhibited tumour growth *in vitro* and *in vivo* by increasing the production of the cholesterol metabolite 24OHC, thus disturbing cholesterol homeostasis in GBM. Moreover, restoration of the activity of the CYP46A1/24OHC signalling axis by EFV, a medication currently in widespread use for the treatment of AIDs, might be a promising GBM therapeutic strategy.

# Materials and Methods

### Cell culture

Patient-derived glioma stem cells (GSCs), GBM#P3, GBM#BG5 and GBM#BG7, were established from GBM surgical specimens at the K. G. Jebsen Brain Tumour Research Centre, Department of Biomedicine, University of Bergen (Bergen, Norway). Cells derived from primary tumours were validated as GSCs in neurosphere formation assays and through the expression of GSC markers, such as SOX2 and OLIG2. Short tandem repeat (STR) profiling (Appendix Table S2) was carried out to confirm GSC identity. The cells were cultured in Neurobasal™ Medium (Gibco/Thermo Fisher Scientific; Waltham, MA) supplemented with B27 supplement (Invitrogen; Carlsbad, CA), 20 ng/ml bFGF (PeproTech; Rocky Hill, NJ), 20 ng/ml EGF (PeproTech) and 5 μg/ml LDL (Millipore, LP2-2MG). Accutase (Thermo Fisher Scientific; Waltham, MA) was used to digest tumorspheres for expansion of GSCs. The serum-grown GBM cell lines (LN229, U251 and LN18) were purchased from the American Type Culture Collection (Manassas, VA) and cultured in Dulbecco's modified Eagle's medium (DMEM; Thermo Fisher Scientific) supplemented with 10% foetal bovine serum (FBS; Clark, VA). Normal human astrocytes (NHAs) were obtained from Lonza (Walkersville, MD) and cultured in the provided astrocyte growth media supplemented with rhEGF, insulin, ascorbic acid, GA-1000, L-glutamine and 5% FBS.

### Ectopic expression

Lentivirus expressing CYP46A1 or a control sequence was obtained from GeneChem (Shanghai, China). A lentivirus with luciferase was obtained from Obio (Shanghai, China). Lentivirus expressing CYP46A1 was used to infect LN229 and GBM#P3 followed by puromycin selection (Sigma-Aldrich; St. Louis, MO) to generate cell lines stably expressing CYP46A1. The ORF of SREBP1 in pEnter and

vector plasmid control were purchased from Vigene Biosciences (Jinan, Shandong, China). Transient transfection was carried out using Lipofectamine 3000 (Life Technologies; Gaithersburg, MD) according to the manufacturer's protocol.

### RNA isolation and quantitative real-time PCR

Total RNA was extracted from cells using TRIzol reagent (Invitrogen; Carlsbad, CA, USA) and reverse-transcribed using the Rever Tra Ace qPCR RT Kit (Toyobo; Osaka, Japan). cDNA was amplified using SYBR Green on the Roche Light Cycler 480 for quantification. The relative expression levels of mRNA were normalized to glyceraldehyde-3-phosphate dehydrogenase (GAPDH). Sequences of the primers used are shown in Appendix Table S3.

### Clinical specimens

Archived paraffin-embedded glioma tissues (WHO grades II-IV) were collected from patients ($n = 58$) who underwent surgery in the Department of Neurosurgery, Qilu Hospital of Shandong University. Normal brain tissue samples ($n = 6$) were taken from trauma patients who underwent partial resection of normal brain as decompression treatment for severe head injuries. The informed consent was obtained from all subjects and that the experiments conformed to the principles set out in the WMA Declaration of Helsinki and the Department of Health and Human Services Belmont Report.

### Immunohistochemistry

Paraffin-embedded blocks containing tissue from traumatic brain injury, glioma tissues or xenografts were sectioned (5 μm). Slides were incubated with primary antibody, rabbit anti-CYP46A1 (ab198889, 1:100, Abcam; Burlingame, CA), rabbit anti-PCNA (ab29, 1:10,000; Abcam) and rabbit anti-cleaved caspase-3 (ab2302, 1:200; Abcam). Detailed protocols have been described previously in Supplementary Materials (Han *et al*, 2018). Staining was evaluated independently by two experienced neuropathologists.

### Western blotting

Cells were lysed with RIPA lysis buffer (Thermo Fisher Scientific; Waltham, MA, USA) supplemented with a protease and phosphatase inhibitor cocktail (Sigma-Aldrich; St. Louis, MO, USA). The BCA Protein Assay Kit (Beyotime, Shanghai, China) was used to determine protein concentration according to the manufacturer's instructions. Equal amounts of protein extracts were separated by SDS–PAGE followed by electrotransfer of protein onto PVDF membrane (Merck Millipore; Billerica, MA, USA). The membrane was blocked with skim milk and incubated with primary antibodies followed by incubation with a horseradish peroxidase (HRP)-conjugated secondary antibody (ZSGB-BIO; Beijing, China) for 1 h. Immunoreactivity was detected with ECL (Merck Millipore; Billerica, MA, USA), and GAPDH was used as a loading control. The following primary antibodies were used: rabbit anti-ABCA1 (ab7360, 1:500, Abcam), rabbit anti-LDLR (ab52818, 1:500, Abcam), rabbit anti-CYP46A1 (ab198889, 1:200, Abcam), rabbit anti-PCNA (ab92552, 1:5,000, Abcam), rabbit anti-cleaved caspase-3 (ab2302, 1:500, Abcam), rabbit anti-OLIG2 (ab109186, 1:2,000, Abcam),

rabbit anti-GFAP (ab33922, 1:2,000, Abcam) and rabbit anti-SREBP1 (ab3259, 1:200, Abcam); rabbit anti-GAPDH (#5174, 1:1,000, Cell Signaling Technology; Danvers, MA), rabbit anti-cleaved PARP (#5625, 1:1,000, Cell Signaling Technology) and rabbit anti-SOX2 (#3579, 1:1,000, Cell Signaling Technology).

## Cell survival and $IC_{50}$ analysis

Cells were seeded in triplicate in 96-well culture plates at 3000 cells per well. Cholesterol (HY-N0322, MedChemExpress; Monmouth Junction, NJ) was dissolved in 40% methyl-β-cyclodextrin (MβCD, HY-101461, MedChemExpress). Corresponding reagents such as 24OHC (HY-N2370, MedChemExpress) and efavirenz (EFV; HY-10572, MedChemExpress) were added to cells in DMEM containing 1% FBS and 1% penicillin/streptomycin for established cell lines or neurobasal medium for patient-derived GSCs. Cell viability assays were performed using the trypan blue assay (15250061, Gibco/Thermo Fisher Scientific) or the Cell Counting Kit-8 (CK04, Dojindo; Rockville, MD) according to the manufacturer's protocol. Raw OD values at 450 nm were analysed using PRISM software for CCK-8. For $IC_{50}$ analysis, EFV or 24OHC concentrations were log10-transformed, and values were normalized to per cent viability relative to vehicle-treated cells.

For the IncuCyte proliferation assays, cells were treated with DMSO, 24OHC or EFV at various concentrations. Cells were monitored and imaged using an IncuCyte® S3 Live-Cell Analysis System (Essen BioScience, Ltd.; Welwyn Garden City, Hertfordshire, UK), and data were analysed by IncuCyte Confluence version 1.5 software (Essen Bioscience). All experiments were performed in triplicate.

## Establishment of intracranial GBM xenografts and efavirenz treatment

Experimental procedures were approved by the Research Ethics Committee of Shandong University and the Ethics Committee of Qilu Hospital (Shandong, China). The Institutional Animal Care and Use Committee (IACUC) of Shandong University approved all animal procedures. Four-week-old male athymic nude mice (Foxn1[nu] mut/mut; SLAC Laboratory Animal Center; Shanghai, China) were bred under specific pathogen-free conditions at 24°C on a 12-h day–night cycle. For orthotopic transplantation, mice were grouped randomly ($n = 5$ per group) and intracranially injected with $1 \times 10^6$ LN229 or GBM#P3 cells resuspended in 10 μl PBS. At day 7 after cell implantation, mice were given drugs (efavirenz) or PBS by oral gavage at 0.1 mg/kg/day until sacrifice based on tumour burden defined by the observation endpoints of the experiments (neurological signs became apparent). GBM#P3 expressing luciferase xenograft growth was monitored with bioluminescent imaging using an *In Vivo* Imaging System (IVIS) Spectrum (Perkin-Elmer; Waltham, MA). Brains were collected and fixed in 4% formaldehyde for haematoxylin and eosin (HE) staining and IHC analysis.

## LC-MS analysis

24OHC measurements were performed by Biotree Biotechnology Co., Ltd (Shanghai, China). Cell samples were prepared as follows: 1,000 μl of extract solvent (precooled at −20°C,

acetonitrile–methanol–water, 2:2:1) was added to pelleted cells, and samples were vortexed for 30 s and homogenized at 45 Hz for 4 min. Media samples were prepared as follows: a 200 μl aliquot of each individual sample was precisely transferred to an Eppendorf tube. Methanol (400 μl) and acetonitrile (400 μl) were added to the aliquot, and the samples were vortexed for 30 s. Cell samples or media were sonicated for 15 min in an ice-water bath followed by incubation at −20°C for 1 h and centrifugation at 13,523 *g* at 4°C for 15 min. A 750 μl aliquot of the clear supernatant was transferred to a new Eppendorf tube and dried under a gentle nitrogen flow. The residual was reconstituted with 150 μl of solution (acetonitrile–methanol–water, 2:2:1), centrifuged at 13,523 *g* at 4°C for 15 min. An 80 μl aliquot of the clear supernatant was transferred to an auto-sampler vial for UHPLC-MS/MS analysis. Stock solutions were individually prepared by dissolving or diluting each standard substance to give a final concentration of 10 mmol/L. The UHPLC separation was carried out using an Agilent 1290 Infinity II Series UHPLC System (Agilent Technologies; Santa Clara, CA), equipped with an Agilent ZORBAX SB-C18 (50 × 2.1 mm, 1.8 μm, Agilent Technologies). Calibration solutions were subjected to UHPLC-MRM-MS/MS analysis using the methods described above. The calibration standard solution was diluted stepwise, with a dilution factor of 2. These standard solutions were subjected to UHPLC-MRM-MS analysis. The signal-to-noise ratios (S/N) were used to determine the lower limits of detection (LLODs) and lower limits of quantitation (LLOQs). The LLODs and LLOQs were defined as the analyte concentrations that led to peaks with signal-to-noise ratios (S/N) of 3 and 10, respectively, according to the US FDA guideline for bioanalytical method validation. The precision of the quantitation was measured as the relative standard deviation (RSD), determined by injecting analytical replicates of a QC sample. The accuracy of quantitation was measured as the analytical recovery of the QC sample determined. The per cent recovery was calculated as [(mean observed concentration)/(spiked concentration)] × 100%. Finally, 24OHC content for each sample was quantified and normalized to whole-cell protein or cell count. Values shown were normalized to controls.

## Cholesterol measurements

Cholesterol content was measured using the Cholesterol Quantitation Kit (Sigma-Aldrich, MAK043-1KT), and filipin staining was performed according to the manufacturer's protocols. For filipin staining (free cholesterol marker), GBM cells were treated for 72 h with 24OHC, EFV (MedChem) or vehicle control. Cells were stained with 0.5 mg/ml filipin in 1% BSA/PBS for 1 h after fixation with 4% paraformaldehyde for 15 min and rinsed 3 × 5 min with PBS. Images were captured by confocal microscopy (Leica TCS SP8; Wetzlar, Germany).

## RNA sequencing

GBM#P3 cells were treated with 20 μM 24OHC or DMSO for 72 h. Cells were harvested, and total RNA was isolated using TRIzol (Invitrogen). RNA sequencing (RNA-seq) was carried out by OE Biotech. Co., Ltd. (Qingdao, Shandong, China). The genome-wide transcriptomic analysis was performed on a set of 3 separate experiments (24OHC and vehicle control treatment groups). *P*-values were

adjusted using Benjamini and Hochberg's multiple test correction procedures. Differential expression was determined based on fold change (FC) and FDR with $|log2(FC)| > 1$ and FDR < 0.01. DAVID (https://david.ncifcrf.gov/) was used to perform GO and KEGG analysis, and data visualization was accomplished using R software. Gene set enrichment analysis (GSEA; https://www.broadinstitute.org/gsea/index.jsp) was performed to find differential phenotypes between 24OHC and vehicle control treatment groups.

## Chromatin immunoprecipitation (ChIP) assays

The EZ-ChIP Immunoprecipitation Kit (Millipore; Billerica, MA, USA) was used to perform ChIP assays. The following antibodies were used: anti-H3K27ac (ab4729, 1:100, Abcam), anti-H3K4me3 (ab8580, 1:100, Abcam) and normal rabbit IgG (#2729, 1:100, Cell Signaling Technology). The sequences of the primers for H3K27ac and H3K4me3 binding sites at the CYP46A1 promoter were the following: F 5′-GTGGTCCACGTGGTACTTCT-3′ and R 5′-CATCCAC GACATCTCGCACA-3′. Briefly, GBM cells or NHAs were cross-linked with 1% formaldehyde solution for 10 min and quenched with 0.125 M glycine. Cells were spun down, washed, resuspended, lysed and ultra-sonicated. Fragmented chromatin extract was precleared with agarose beads from the ChIP Kit and incubated overnight with antibodies or normal rabbit IgG as the control. After washing, elution and reverse cross-linking, DNA was analysed by qPCR.

## Flow cytometry

Cells were harvested, rinsed twice with PBS, fixed with 75% (v/v) ethanol overnight at 4°C and stained with propidium iodide (PI; BD Biosciences; San Jose, CA, USA) at room temperature for 20 min. To detect apoptosis, cells were rinsed with PBS, resuspended in 200 μl binding buffer and incubated with Annexin V-FITC and PI (BD Biosciences; San Jose, CA, USA) at room temperature for 15 min. Both cell cycle distribution and apoptosis were analysed on a C6 flow cytometer (BD Biosciences; San Jose, CA, USA). Data were postprocessed using NovoExpress Software (ACEA Biosciences; San Diego, California, USA).

## Caspase-3/7 activity assay

*In vitro* caspase-3/7 activity was determined using CellEvent™ Caspase-3/7 Green Detection Reagent (Invitrogen; Carlsbad, CA, USA) according to the manufacturer's protocol. For the kinetic assay, LN229 cells were treated with 24OHC at different concentrations (0–20 μM) and 2 μM Caspase-3/7 Green Detection Reagent. IncuCyte® S3 Live-Cell Analysis System (Essen BioScience, Ltd.; Welwyn Garden City, Hertfordshire, UK) was used to visualize the progression of apoptosis using FITC/Alexa Fluor™ 488 filter settings at desired time points. Cells were monitored and imaged with the IncuCyte® S3 Live-Cell Analysis System (Essen BioScience, Ltd.; Welwyn Garden City, Hertfordshire, UK), and graphics were generated with IncuCyte software.

## Tumorsphere formation assay

GSCs (1,000 cells/ml/well) were seeded in 6-well ultra-low adhesion plates (Corning Inc.; Corning, NY, USA) or general plates (Corning Inc.) and cultured for 10 days. Inverted phase-contrast microscopy (Nikon; Tokyo, Japan) was used to count and acquire images of the non-adherent tumorspheres.

## Colony formation assay

LN18 or LN229 (1,000 cells/well) cells were seeded in 6-well culture plates in triplicate and incubated at 37°C in a humidified chamber of 5% $CO_2$ for 14 days. Colonies were washed with PBS, fixed in methanol for 15 min and stained with crystal violet for 15 min at room temperature. Images of colonies were acquired under bright-field microscopy (Nikon; Tokyo, Japan), and those with a diameter of ≥ 100 μm were counted.

## Live–dead cell staining

Live–dead cell staining was performed using the Live/Dead Cell Double Staining Kit (Sigma-Aldrich; St. Louis, MO, USA) according to manufacturer's protocol. Briefly, brain organoids were treated with DMSO or efavirenz for 3 days. The medium was changed before live–dead cell staining to exclude the influence of DMSO and efavirenz on staining. Calcein-AM and PI working solutions were prepared in PBS at a proper dilution. The staining solutions were mixed into a working solution with the brain organoid suspension at a ratio of 1:2 (v/v) and incubated at 37°C for 15 min. Images for live–dead cells were captured with confocal microscopy (Leica TCS SP8; Leica Microsystems; Wetzlar, Germany).

## Brain organoid cultures

The preparation and culture of brain organoids have been described previously (Bjerkvig *et al*, 1986). Briefly, rat foetal brains at the 18th day of gestation were dissected out aseptically, cut into 0.5 × 0.5 mm pieces and rinsed three times with PBS. Single cells were obtained through serial trypsinization and seeded into agar-coated multi-well dishes for 21 days to form mature brain organoids. Images were captured, after treatment of brain organoids with 24OHC, EFV or DMSO (vehicle control) for 3 days, using bright-field microscopy (Nikon; Tokyo, Japan).

## Immunofluorescence

GBM#P3 tumorspheres were seeded on poly-D-lysine (Sigma-Aldrich)-coated coverslips. Cells were fixed in 4% paraformaldehyde in PBS and blocked with 10% goat serum (Beyotime, Shanghai, China) and 0.3% Triton X-100 in PBS for 30 min. Slides were incubated with mouse monoclonal SOX2 antibody (sc-365823, 1:100, Santa Cruz Biotechnology; Dallas, TX) overnight and incubated with anti-mouse IgG-conjugated Alexa Fluor 568 (A-11031, 1:500, Invitrogen) for 2 h at room temperature. Nuclei were counterstained with DAPI (C1005, Beyotime; Haimen, Jiangsu, China). Images were captured with a Leica inverted microscope.

## Statistical analysis

Kaplan–Meier survival curves were generated and compared using the log-rank test. The association between *CYP46A1* expression and

## The paper explained

### Problem

Glioblastoma (GBM) is the most common primary malignant brain tumour in adults. Despite aggressive therapy, including maximal surgical resection followed by radiotherapy and temozolomide treatment, the median patient survival is 14.6 months from initial diagnosis. Emerging evidence indicates that dysregulated cholesterol metabolism is a hallmark of many cancers including glioblastoma. However, the precise role of cholesterol metabolism in growth and progression of glioblastoma and its therapeutic potential have not yet been elucidated. Here, we analysed cholesterol homeostasis-related genes systematically in glioblastoma databases and identified a functional role of cholesterol 24-hydroxylase (CYP46A1) in glioblastoma growth and progression.

### Results

We conducted a large-scale *in silico* analysis of publicly available genomic datasets of gliomas. Using this approach, we identified the gene CYP46A1 as one of the most dramatically down-regulated genes involved in GBM cholesterol homeostasis. CYP46A1 is a brain-specific enzyme responsible for the elimination of cholesterol through the conversion of cholesterol into 24(S)-hydroxycholesterol (24OHC). Mechanistically, ectopic expression of CYP46A1 suppressed glioma stem cell proliferation and *in vivo* tumour growth by increasing 24OHC, which led to a decrease in GBM cholesterol levels. These results led us to examine the blood–brain barrier permeable anti-HIV drug, efavirenz (EFV), which is known to activate CYP46A1, for the treatment of GBM. EFV increased 24OHC levels in GBM cells and suppressed GBM cell growth.

### Impact

Our results provide evidence that drug repurposing of EFV targeting dysregulated cholesterol metabolism might be a promising therapeutic strategy for glioblastoma.

clinicopathological characteristics was determined in a two-tailed chi-square test. For animal studies, mice were randomly allocated to control or treated group. No blinding was done. The Kolmogorov–Smirnov test was used to assess the normal distribution of data. Statistical analysis was performed using an unpaired Student's *t*-test for two-group comparison and one-way analysis of variance (ANOVA) for multi-group comparisons using GraphPad Prism 7.0 (La Jolla, CA). All the experiments were repeated at least three times with triplicates unless stated otherwise. Data for each treatment group were represented as the mean ± SEM. All tests were two-sided, and *P*-values < 0.05 were considered to be statistically significant.

## Data availability

The datasets in this study are available in the following databases:

The RNA-seq data: Gene Expression Omnibus (GEO) with the record numbers GSE54047 (https://www.ncbi.nlm.nih.gov/geo/query/acc.cgi?acc=GSE54047), GSE46016 (https://www.ncbi.nlm.nih.gov/geo/query/acc.cgi?acc=GSE46016) and GSE46014 (https://www.ncbi.nlm.nih.gov/geo/query/acc.cgi?acc=GSE46014).

**Expanded View** for this article is available online.

## Acknowledgements

We thank Dr. Justin Vareecal Joseph for establishing and providing primary GBM cells. This work was supported by the National Natural Science Foundation of China (81972351, 81702474 and 81702475), the Department of Science & Technology of Shandong Province (2017CXGC1502, 2018CXGC1503 and 2018GSF118082), the Special Foundation for Taishan Scholars (ts20110814, tshw201502056 and ts201511093), the Shandong Provincial Natural Science Foundation (ZR2017MH116), the China Postdoctoral Science Foundation (2018M642666), the Jinan Science and Technology Bureau of Shandong Province (201704096), the Norwegian Cancer Society, the Norwegian Research Council (ES563961), Haukeland University Hospital, Helse-Vest and the University of Bergen.

## Author contributions

MH, XL, JW and RB conceived the project, designed the experiments and wrote the paper; MH, SW, NY, WZ and XW conducted the *in vitro* and *in vivo* experiments; BH and AC collected patient samples; MH and SW performed IHC analyses; and FT, HSS, GL and TD provided the bioinformatics and statistical support, and HM performed additional neuropathological analyses.

## Conflict of interest

The authors declare that they have no conflict of interest.

## For more information

(i)   https://www.cbioportal.org/Cbioportal
(ii)  http://gepia.cancer-pku.cn/index.html
(iii) https://www.proteinatlas.org/

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
