## [Review Process File · EMBO Molecular Medicine]

Therapeutic implications of altered cholesterol homeostasis mediated by loss of CYP46A1 in human glioblastoma

Mingzhi Han, Shuai Wang, Ning Yang, Xu Wang, Wenbo Zhao, Halala Sdik Saed, Thomas Daubon, Bin Huang, Anjing Chen, Gang Li, Hrvoje Miletic, Frits Thorsen, Rolf Bjerkvig, Xingang Li, Jian Wang

Review timeline:

Submission date:	22 May 2019
Editorial Decision:	17 June 2019
Revision received:	12 September 2019
Editorial Decision:	10 October 2019
Revision received:	19 October 2019
Accepted:	23 October 2019

Editor: Céline Carret

Transaction Report:

1st Editorial Decision

17 June 2019

Thank you for the submission of your manuscript to EMBO Molecular Medicine. We have now heard back from the three referees whom we asked to evaluate your manuscript.

I am happy to report that the three referees are positive and supportive of publication. A few additional experiments should be performed, along with the provision of clarifications and details to increase the conclusiveness of the findings.

We would therefore welcome the submission of a revised version within three months for further consideration and would like to encourage you to address all the criticisms raised as suggested to improve conclusiveness and clarity. Please note that EMBO Molecular Medicine strongly supports a single round of revision and that, as acceptance or rejection of the manuscript will depend on another round of review, your responses should be as complete as possible.

I look forward to receiving your revised manuscript.

***** Reviewer's comments *****

Referee #1 (Remarks for Author):

In this Manuscript, the Authors enlighten the role of the enzyme CYP46A1 in gliomagenesis and its therapeutic potential as druggable target.

They convincingly show that CYP46A1, that converts cholesterol in 24(S)-hydroxycholesterol (24OHC), is decreased in GBM samples compared to normal brain tissue. Applying a combination of in vitro and in vivo experiments, plus omics, they demonstrate that its ectopic expression impairs GBM tumorigenic properties by increasing 24OHC levels, that in turns decreases cholesterol levels and regulates LXR and SREBP signaling.

The cellular models employed, both differentiated cell lines and primary Glioblastoma Stem Cells, is adequate, because demonstrate that the CYP46A1/24OHC axis is important to promote survival and tumorigenic properties of both differentiated and stem cell like compartment of GBM.

The dissection of the molecular mechanisms underlying the observed effects is supported by a good amount of rescue experiments.

The translational value of these results is increased by the possibility to target GBM using Efavirenz (EFV), whose ability to cross the blood-brain barrier has been already demonstrated.

Overall, the quality of this manuscript is high, and only a minor revision is required.

-Fig.2C-4C-5I: growth curve/viability experiment has been conducted using CCK8 kit. This Kit infers the number of viable cells through a colorimetric reaction that reflects the activity of mitochondrial dehydrogenases. Being a metabolic assay, this may be influenced by the changes of the metabolic status of GBM cells that can eventually occur upon the administration of 24HO or treatment able to interfere with cholesterol metabolism. To this possible bias, these experiments should be repeated measuring the number of viable/dead cells by manual counting with an appropriate staining, such as Erythrosine or Trypan Blue.

-FigS5A-B: Plot for H3K27ac and H4K4me3 are shown, but Chip-seq analysis should be showed in a more quantitative manner. For instances, a scale for the number of reads count must be indicated in the peaks plot.

-FigS5A-B: the validation of these results on the samples used by Authors should be exploited by ChIP-qPCR.

Referee #2 (Comments on Novelty/Model System for Author):

Overall, the study is very well designed and executed. One weakness is the use of stable cells lines (U251 and LN229 in majority of the assays despite the fact that the group had access to primary cells/PDX models, which are superior to stable cell lines.

Referee #2 (Remarks for Author):

The manuscript by Han et al. entitled 'Therapeutic implications of altered cholesterol homeostasis mediated by loss of CYP46A1 in human glioblastoma' identifies CYP46A1 hydrolase as a driver of deregulated cholesterol metabolism in glioblastoma cells. Even though the concept of glioblastoma dependency on cholesterol and its role in gliomagenesis is not novel, this study provides an additional (and important) insight into the mechanism underpinning this dependency. This is an elegant and well-designed study with high clinical relevance. Minor weakness is the use of stable cell lines (U251 and LN229), which is rather surprising as the group has access to primary cell cultures and PDX models, which are superior to stable cell lines.

1. The authors examined the active enhancer landscape of CYP46A1 across 3 matched cancer stem-like cells (GSCs) and their differentiated counterparts (DGCs), showing decrease in GSCs - how significant (statistical analysis missing) is this difference? - does this difference translate into differential CYP46A1 protein expression?
2. H3K27ace-ChIPseq showed 'decreased transcription' of CYP46A1? The authors should validate this using luciferase reporter assay. Is the promoter methylated in glioma cells (GSCs and DGCs)?
3. Is there any difference between CSCs versus DGCs in regards to their dependency on exogenous cholesterol and/or sensitivity to 24OHC/EFV (viability assay)?
4. What is the effect of 24OHC/EFV on non-malignant brain control - such as normal human

astrocytes?

5. IVY GBM data set analysis revealed higher CYP46A1 expression in the leading edge. Cholesterol has been shown to impact membrane fluidity - is the loss of CYP46A1 associated with deregulated membrane repair, thus impacting migration and/or invasion (in vitro and in vivo)?

Minor:

1. p.5 - patient derived GSCs were validated by a series of functional assays - please, specify which assays and perhaps include in supplementary file
2. Schematic fig.7 lacks symbol explanation (what is the difference between orange and yellow circles for example?) and is a bit hard to follow based on the corresponding figure legend

Referee #3 (Comments on Novelty/Model System for Author):

The technical quality of the experiments is high and the statistical analysis adequate. Materials and methods could be more detailed.

Novelty is medium in my opinion, as a large body of the findings on cholesterol metabolism as a specific vulnerability in GBM has been described before (but correctly cited and discussed by the authors in the context of their work). However, the link between CYP46A1, and 24-OHC as the specific metabolite has never been demonstrated experimentally.

Still I feel that the medical impact is high, as the authors demonstrate the feasibility of pharmacological activation of CYP46A1 for treatment of GBM (with an already approved drug). However, I would like to see the experiments ruling out off-target effects, as detailed in the comments to the authors.

Referee #3 (Remarks for Author):

The manuscript by Han et al. analyses the role of the enzyme CYP46A1, a cholesterol 24-hydroxylase in glioblastoma multiforme (GBM). It has previously been shown, that cholesterol metabolism is a critical vulnerability of GBM and that suppressing cholesterol uptake can induce cell death in GBM cells and suppress tumor growth in vivo. Metabolites of cholesterol (hydroxycholesterols) have been shown to act as ligands for the nuclear receptor LXR, activation of which in turn leads to reduced intracellular cholesterol levels by induction of cholesterol efflux and repression of cholesterol uptake. It has previously been noted, that the levels of hydroxycholesterols, and concomitantly LXR activation, are significantly reduced in GBM as compared to normal human astrocytes. In line with those findings, it has been shown that an agonist of LXR could be an effective treatment for GBM (Villa et al. Cancer Cell 2016). In the brain, CYP46A1 is the major enzyme responsible for generating 24-hydroxycholesterol (24-OHC) from cholesterol and Villa et al. already noted that is significantly downregulated in GBM, suggesting a link between CYP46A1 and reduced 24-OHC levels in the brain. This link however has to my knowledge never been addressed experimentally.

Han et al. investigate the contribution of reduced CYP46A levels to the enhanced cholesterol uptake in GBM and identify activation of CYP46A1 as a druggable vulnerability in GBM. The main findings of the manuscript are the confirmation that CYP46A1 is downregulated in a variety of GBM patient datasets, correlates with aggressiveness and survival in GBM patients. The authors then further show that increased expression of CYP46A1 leads to reduced GBM growth and increased survival in vitro and in vivo, respectively. Han et al. further demonstrate that CYP46A1 acts via 24-OHC in repressing GBM growth and that its effects are mediated by LXR and SREBP1, suppressing intracellular cholesterol levels. Finally, they demonstrate that pharmacological stimulation of CYP46A1 activity could be a therapeutic strategy for treatment of GBM.

While alteration of the cholesterol axis as a target for GBM has previously been described and thus part of the experiments feels like a reproduction of data from Villa et al., I am of the opinion that this is an important manuscript. First, while the involvement of CYP46A1 downregulation in the altered cholesterol metabolism (and thus 24-OHC as a key player) in GBM has been strongly

suggested this link has never been directly demonstrated. Second, the authors show that pharmacological alteration of CYP46A1 activity could provide a potential therapeutic strategy with direct implications for treatment of GBM, especially as the drug used is already approved.

In general, the manuscript is well written and the conclusions are largely supported by the experimental data. I have only a few specific comments and questions:

1: In the experiments involving the lentiviral expression of CYP46A1 (Figs. 2 and 3) it would be valuable to get a better idea of CYP46A levels between the different cell lines, GBM of different grades and normal human astrocytes. As each cell line/qPCR is provided in a sparse panel this is impossible. A comparison across different materials would also really help to judge the levels of overexpression (do they reach normal astrocyte levels?).

2: On page 10 the authors state 'Up-regulation of CYP46A1 in the neural GB; subtype (...)'. Is this really an upregulation or rather a less pronounced downregulation as compared to normal astrocytes?

3: In Fig. 4 the authors show rescue experiments where Cholesterol addition can partially rescue the effects of 24-OHC. However, the culture medium already contains cholesterol, either added via LDL for GSC cultures or naturally present in the FBS. How does the additional amount of cholesterol added compare to the already present cholesterol? Would a rescue effect be expected at all?

4: In Fig. 6D, titration of efavirenz to concentrations exceeding 50 μ M has an increased effect in suppressing cell viability. However, according to Anderson et al. (JBC2016, cited by the authors) concentrations of 60 μ M and higher actually reduced CYP46A1 activity. While the data from Anderson are in vitro enzyme assays this still leaves the possibility that the effect of efavirenz seen by the authors might be an off target effect. One way to assess this would be to downregulate or delete CYP46A1 in the cell lines by shRNA or CRISPR/Cas9. In those lines, efavirenz should have no effect. I feel this would be an experiment that could significantly strengthen the message of the manuscript that an already approved drug could be effective for treatment of GBM.

5: The authors show that SREBP1 activity is not regulated on the level of its proteolytic cleavage, but appears rather to be on the transcriptional level. Do the authors have any suggestion or potential candidate pathways for this regulation?

6: Materials and Methods: The authors should give more specific information on some of the reagents, such as catalog number/clone number when the description is ambiguous. I noted this especially for the antibodies used and reagents such as LDL (there are several types available from Millipore).

1st Revision - authors' response

12 September 2019

Referee #1 (Remarks for Author):

In this Manuscript, the Authors enlighten the role of the enzyme CYP46A1 in gliomagenesis and its therapeutic potential as druggable target.

They convincingly show that CYP46A1, that converts cholesterol in 24(S)-hydroxycholesterol (24OHC), is decreased in GBM samples compared to normal brain tissue. Applying a combination of in vitro and in vivo experiments, plus omics, they demonstrate that its ectopic expression impairs GBM tumorigenic properties by increasing 24OHC levels, that in turns decreases cholesterol levels and regulates LXR and SREBP signalling.

The cellular models employed, both differentiated cell lines and primary Glioblastoma Stem Cells, is adequate, because demonstrate that the CYP46A1/24OHC axis is important to promote survival and tumorigenic properties of both differentiated and stem cell like compartment of GBM. The dissection of the molecular mechanisms underlying the observed effects is supported by a good amount of rescue experiments.

The translational value of these results is increased by the possibility to target GBM using Efavirenz (EFV), whose ability to cross the blood-brain barrier has been already demonstrated.

Overall, the quality of this manuscript is high, and only a minor revision is required.

1. Fig 2C-4C-5I: growth curve/viability experiment has been conducted using CCK8 kit. This Kit infers the number of viable cells through a colorimetric reaction that reflects the activity of mitochondrial dehydrogenases. Being a metabolic assay, this may be influenced by the changes of the metabolic status of GBM cells that can eventually occur upon the administration of 24HO or treatment able to interfere with cholesterol metabolism. To this possible bias, these experiments should be repeated measuring the number of viable/dead cells by manual counting with an appropriate staining, such as Erythrosine or Trypan Blue.

Response: We agree that CCK-8 assay may be influenced by changes in the metabolic status of cells. As suggested by the referee, we have now performed new experiments measuring the number of viable cells by manual counting using the Trypan Blue test. The results were consistent with our previous CCK-8 data. Accordingly, we have modified the figures (Fig. 2C, 4C, and 5I) in the revised manuscript.

2. FigS5A-B: Plot for H3K27ac and H4K4me3 are shown, but Chip-seq analysis should be showed in a more quantitative manner. For instances, a scale for the number of reads count must be indicated in the peaks plot.

Response: Following reviewer's advice, we have now included the scale in the revised ChIP-seq plot (Fig. S3A and S3B). We also performed ChIP-qPCR to validate the results from ChIP-seq and quantified histone modification changes (please see also Question 3).

3. FigS5A-B: the validation of these results on the samples used by Authors should be exploited by ChIP-qPCR.

Response: As the reviewer suggested, we have now performed ChIP-qPCR to validate the ChIP-seq data. We compared the H3K27ac levels in the *CYP46A1* promoter between normal human astrocytes (NHA) and primary GBM cells. The results showed that H3K27ac levels in GBM#P3 were decreased compared to levels in NHA (See figure below). We also observed that both H3K4me3 and H3K27ac levels were decreased in GBM#P3 compared to serum-induced differentiated GBM#P3 cells (DGC was induced with DMEM containing 1% FBS for 7 days) using ChIP-qPCR [Unpublished data removed upon the authors' request]. Taken together, these data indicate that histone modifications may partially explain reduced *CYP46A1* expression in GBM. These data are now shown in Fig. S3D.

Referee #2 (Comments on Novelty/Model System for Author):

Overall, the study is very well designed and executed. One weakness is the use of stable cells lines (U251 and LN229 in majority of the assays despite the fact that the group had access to primary cells/PDX models, which are superior to stable cell lines.

Response: We acknowledge that this may be a minor weakness, yet to perform and redo all the experiments that are included in our extensive work, using cells/PDX models, would imply elaborate work that most likely will not lead to other conclusions related to the results presented. For instance, it should be emphasized that most results obtained, using U251 and LN229, were validated in human derived cells/PDX models (GBM#P3) (See Fig 2, 3, 4 and 6). It should in this context also be mentioned that we have previously shown that GBM#P3 has DNA copy number variations that reflect human GBMs (Keunen et al, 2011).

Referee #2 (Remarks for Author):

The manuscript by Han et al. entitled 'Therapeutic implications of altered cholesterol homeostasis mediated by loss of *CYP46A1* in human glioblastoma' identifies *CYP46A1* hydrolase as a driver of deregulated cholesterol metabolism in glioblastoma cells. Even though the concept of glioblastoma dependency on cholesterol and its role in gliomagenesis is not novel, this study provides an additional (and important) insight into the mechanism underpinning this dependency. This is an

elegant and well-designed study with high clinical relevance. Minor weakness is the use of stable cell lines (U251 and LN229), which is rather surprising as the group has access to primary cell cultures and PDX models, which are superior to stable cell lines.

Response: We thank the reviewer for this very positive feed-back to our work. Regarding the comment about the cell lines used, see above.

1. The authors examined the active enhancer landscape of CYP46A1 across 3 matched cancer stem-like cells (GSCs) and their differentiated counterparts (DGCs), showing decrease in GSCs - how significant (statistical analysis missing) is this difference? - does this difference translate into differential CYP46A1 protein expression?

Response: We thank the reviewer for this comment and accordingly, -we have now included in our revised manuscript a scale in the ChIP-seq data figure (Fig. S3A-S3B) and statistically validated the results using ChIP-qPCR (Fig. S3D). Moreover, as shown in Fig S3E [Unpublished data removed upon the authors' request], we performed western blot analysis to determine the differences in CYP46A1 protein levels. GSC cells showed lower expression of CYP46A1 relative to that in DGC cells, which was accompanied by a decreased expression of GSC markers (SOX2 and OLIG2) and an induction of the differentiation marker GFAP. We have modified this section in the revised manuscript.

2. H3K27ac-ChIPseq showed 'decreased transcription' of CYP46A1? The authors should validate this using luciferase reporter assay. Is the promoter methylated in glioma cells (GSCs and DGCs)?

Response: We acknowledge this comment. Consequently, in order to validate our ChIP-seq data, we performed H3K4me3- and H3K27ac-ChIP-qPCR at the promoter region of *CYP46A1* to confirm a decreased transcriptional activity in primary GBM cells (Fig. S3D). Moreover, we performed western blots to confirm the decreased translational activity (Fig. S3E). Our data thus confirm that CYP46A1 is frequently decreased in GBM.

Since promoter methylation is another important epigenetic mechanism controlling gene expression in cancer, we further assessed *CYP46A1* promoter methylation in gliomas. Using data extracted from the TCGA datasets, we found no significant differences in promoter methylation levels between GBM and non-neoplastic brain tissues ($P > 0.05$; [Unpublished data removed upon the authors' request]). Moreover, we observed no significant correlation between *CYP46A1* mRNA levels and DNA methylation at the *CYP46A1* transcriptional start site (TSS) in both LGG and GBM samples (Figure. B below). This analysis suggests that promoter methylation may not contribute to *CYP46A1* silencing in gliomas.

3. Is there any difference between CSCs versus DGCs in regards to their dependency on exogenous cholesterol and/or sensitivity to 24OHC/EFV (viability assay)?

Response: We thank the reviewer for this insightful comment. Since GSCs and DGCs display different endogenous CYP46A1 expression levels, there could be a difference regarding their dependency on dysregulation of cholesterol homeostasis.

Accordingly, we performed new experiments using GBM#P3 (GSC) and serum-induced differentiated GBM#P3 cells (DGC). We observed that adding methyl- β -cyclodextrin (M β CD; 20 μ M), which is a chemical known to reduce cellular cholesterol content, led to a significant inhibition of GSC cell growth (Trypan Blue staining). This effect was significantly weaker in DGCs ($P < 0.001$). Similarly, GSCs were more sensitive to 24OHC (20 μ M) treatment in comparison to DGCs ($P < 0.001$), while no significant difference was observed under treatment with EFV (20 μ M; $P > 0.05$). In summary, these results show that GSCs have a high dependency on exogenous cholesterol and increased sensitivity to 24OHC. For the reviewer's information, we have included the results below.

4. What is the effect of 24OHC/EFV on non-malignant brain control - such as normal human astrocytes?

Response: This information was already present in our first version where we tested the effect of 24OHC/EFV on normal human astrocytes (NHA) as well as on rat brain organoids. Flow cytometry and Live/Dead cell double staining demonstrated that NHA and rat brain organoids were resistant to

24OHC (20 μ M) treatment (Fig EV. 3A-3C). Flow cytometry also demonstrated that EFV induced apoptosis in GBM cells ($P < 0.001$, Fig. 6G), while sparing NHA and normal brain organoids (Fig. S9A-S9C). In summary, these data show that 24OHC/EFV specifically inhibits GBM growth.

5. IVY GBM data set analysis revealed higher CYP46A1 expression in the leading edge. Cholesterol has been shown to impact membrane fluidity – is the loss of CYP46A1 associated with deregulated membrane repair, thus impacting migration and/or invasion (in vitro and in vivo)?

Response: This is a good question. As CYP46A1/24OHC contributes to an altered cellular cholesterol status, we wondered whether this change could impair cellular membrane structure and cell invasion properties. We therefore performed additional experiments in order to address this issue.

In our present study, we examined the whole transcriptome using RNA-seq to explore mechanisms of downstream of CYP46A1/24OHC. Gene Set Enrichment Analysis (GSEA) was also performed to find phenotypes distinguishing 24OHC treatment from controls. Despite a significant association between 24OHC treatment and the dysregulation of cholesterol homeostasis (Fig. 5D), we observed that 24OHC treatment exhibited no significant enrichment for the signatures “Component of membrane structure” ($P > 0.05$; Figure. A below) or “Membranes raft”, which contain most of membrane cholesterol and influence membrane fluidity ($P > 0.05$). In addition, in a functional study, the 3D spheroid invasion assay, we found that overexpression of CYP46A1 had no impact on the invasive properties of GBM ($P > 0.05$; Figure. B below), suggesting that CYP46A1-mediated cholesterol changes may not contribute to cancer cell invasion. It should also be noted that the IVY RNA-seq project (<http://glioblastoma.alleninstitute.org/>) was performed using histologically-distinct anatomic features and laser microdissection of GBM tissues. The leading edge (LE), identified with H&E staining, is the outermost boundary of the tumour, where the ratio of tumor to normal cells is about 1-3/100 (Puchalski et al, 2018). Thus, the higher CYP46A1 levels observed in the LE are more likely due to its high expression in the adjacent normal brain cells.

Since this is a negative result, we have chosen not to include this information in our revised version.

Minor:

1. p.5 - patient derived GSCs were validated by a series of functional assays - please, specify which assays and perhaps include in supplementary file

Response: Patient-derived GSCs, including GBM#P3, GBM#BG5, and GBM#BG7, were established from GBM surgical specimens at the K. G. Jebsen Brain Tumour Research Centre, Department of Biomedicine, University of Bergen (Bergen, Norway). These cells were validated in the neurosphere formation assay [Unpublished data removed upon the authors' request]. In addition, the expression of GSC markers, such as SOX2 and OLIG2, were examined by western blots (Fig. S3E). Accordingly, we have modified this section in the methods section of our revised manuscript.

2. Schematic fig.7 lacks symbol explanation (what is the difference between orange and yellow circles for example?) and is a bit hard to follow based on the corresponding figure legend

Response: We have now added a better explanation of Fig. 7. The figure legend has been revised accordingly in our revised version:

Figure 7. Schematic representation of CYP46A1 contribution in GBM progression.

(Left panel) GBM cells have an excessive cholesterol accumulation caused by CYP46A1 down regulation, which leads to a reduced 24OHC production that in turn cause a suppression of LXR signaling and an induction of SREBP targets. (Right panel) Restoration of CYP46A1 activity by EFV impairs GBM tumorigenic properties by increasing 24OHC levels, which in turns decreases intracellular cholesterol levels. Mechanistically, EFV treatment leads to the up-regulation of LXR target genes including *ABCA1*, and down-regulation of SREBP1 activity and LDLR expression.

Referee #3 (Comments on Novelty/Model System for Author):

The technical quality of the experiments is high and the statistical analysis adequate. Materials and methods could be more detailed.

Novelty is medium in my opinion, as a large body of the findings on cholesterol metabolism as a specific vulnerability in GBM has been described before (but correctly cited and discussed by the authors in the context of their work). However, the link between CYP46A1, and 24-OHC as the specific metabolite has never been demonstrated experimentally.

Still I feel that the medical impact is high, as the authors demonstrate the feasibility of pharmacological activation of CYP46A1 for treatment of GBM (with an already approved drug). However, I would like to see the experiments ruling out off-target effects, as detailed in the comments to the authors.

Referee #3 (Remarks for Author):

The manuscript by Han et al. analyses the role of the enzyme CYP46A1, a cholesterol 24-hydroxylase in glioblastoma multiforme (GBM). It has previously been shown, that cholesterol metabolism is a critical vulnerability of GBM and that suppressing cholesterol uptake can induce cell death in GBM cells and suppress tumor growth *in vivo*. Metabolites of cholesterol (hydroxycholesterols) have been shown to act as ligands for the nuclear receptor LXR, activation of which in turn leads to reduced intracellular cholesterol levels by induction of cholesterol efflux and repression of cholesterol uptake. It has previously been noted, that the levels of hydroxycholesterols, and concomitantly LXR activation, are significantly reduced in GBM as compared to normal human astrocytes. In line with those findings, it has been shown that an agonist of LXR could be an effective treatment for GBM (Villa et al. Cancer Cell 2016). In the brain, CYP46A1 is the major enzyme responsible for generating 24-hydroxycholesterol (24-OHC) from cholesterol and Villa et al. already noted that is significantly downregulated in GBM, suggesting a link between CYP46A1 and reduced 24-OHC levels in the brain. This link however has to my knowledge never been addressed experimentally.

Han et al. investigate the contribution of reduced CYP46A levels to the enhanced cholesterol uptake in GBM and identify activation of CYP46A1 as a druggable vulnerability in GBM. The main findings of the manuscript are the confirmation that CYP46A1 is downregulated in a variety of GBM patient datasets, correlates with aggressiveness and survival in GBM patients. The authors then further show that increased expression of CYP46A1 leads to reduced GBM growth and increased survival *in vitro* and *in vivo*, respectively. Han et al. further demonstrate that CYP46A1 acts via 24-OHC in repressing GBM growth and that its effects are mediated by LXR and SREBP1, suppressing intracellular cholesterol levels. Finally, they demonstrate that pharmacological stimulation of CYP46A1 activity could be a therapeutic strategy for treatment of GBM.

While alteration of the cholesterol axis as a target for GBM has previously been described and thus part of the experiments feels like a reproduction of data from Villa et al., I am of the opinion that this is an important manuscript. First, while the involvement of CYP46A1 downregulation in the altered cholesterol metabolism (and thus 24-OHC as a key player) in GBM has been strongly suggested this link has never been directly demonstrated. Second, the authors show that pharmacological alteration of CYP46A1 activity could provide a potential therapeutic strategy with direct implications for treatment of GBM, especially as the drug used is already approved.

In general, the manuscript is well written and the conclusions are largely supported by the experimental data. I have only a few specific comments and questions:

1: In the experiments involving the lentiviral expression of CYP46A1 (Figs. 2 and 3) it would be valuable to get a better idea of CYP46A levels between the different cell lines, GBM of different grades and normal human astrocytes. As each cell line/qPCR is provided in a spate panel this is impossible. A comparison across different materials would also really help to judge the levels of overexpression (do they reach normal astrocyte levels?).

Response: We thank the reviewer for this valuable comment. As suggested by the referee, we examined the expression levels of CYP46A1 across different cell lines by western blots. Normal human astrocytes (NHA) displayed abundant CYP46A1 protein levels, while GBM cells (GBM#P3, GBM#05, GBM#BG7, LN229, U251, and LN18) showed much lower expression of the protein (Figure A below). This is consistent with both our *in silico* analysis and immunohistochemistry showing that CYP46A1 is frequently lost in GBM samples (Fig. 1 and Fig. S2A-S2D). In addition,

CYP46A1 levels were also assessed in lenti-CYP46A1 stable expressed GBM cells and NHA (as shown in the Figure B below). These results further validate an increased expression of CYP46A1 in GBM cells transfected with lenti-CYP46A1. Accordingly, we have now included these results in the revised manuscript (Fig. S2E).

2: On page 10 the authors state 'Up-regulation of CYP46A1 in the neural GB; subtype (...)'. Is this really an upregulation or rather a less pronounced downregulation as compared to normal astrocytes?

Response: We thank the reviewer for this valuable comment. We have now modified Fig. 1G and included normal brain samples (see Figure below). Although there is a general down-regulation of *CYP46A1* in GBM compared to normal brain, higher expression of *CYP46A1* was also observed in the Neural GBM molecular subtype, which is associated with a more favorable prognosis, relative to the other subtypes based on the TCGA Verhaak-2010 molecular classification of GBM (Noussmeh et al, 2010).

3: In Fig. 4 the authors show rescue experiments where Cholesterol addition can partially rescue the effects of 24-OHC. However, the culture medium already contains cholesterol, either added via LDL for GSC cultures or naturally present in the FBS. How does the additional amount of cholesterol added compare to the already present cholesterol? Would a rescue effect be expected at all?

Response: The culture conditions in our rescue experiment were established based on previous reports (Guo et al, 2011; Villa et al, 2016), where a lower concentration of FBS (1%) was utilized for an adherent cell functional assay. In serum-free GSC cultures, we added 5 µg/mL LDL to the neurobasal culture medium, which is a lower concentration compared to the culture conditions of cancer stem cells (Pei et al, 2018). As for the concentration of cholesterol for the rescue study, we used 0.5 - 0.75 µg/mL exogenous cholesterol (complexed to methyl-β-cyclodextrin, which enables cholesterol to permeate cells) as reported (Villa et al, 2016).

To further confirm the cholesterol content in our culture system, we used the Cholesterol Quantitation Kit to determine cholesterol concentrations in the basic media. The results showed that the FBS (1%) culture medium contains 4.972 µg/mL cholesterol, which is consistent with a previous publication which reported a cholesterol concentration of 120-630 µg/mL in FBS (stock solution) ((Gstraunthaler, 2003). In the neurobasal culture medium, the cholesterol concentration is 0.474 µg/mL.

In our present study, 24OHC treatment resulted in a significant decrease in total intracellular cholesterol levels in GBM cells (Fig 4A and 4B). However, addition of 0.5 µg/mL exogenous cholesterol to the culture medium partially rescued the inhibitory effect of 24OHC in GBM cells (Fig. 4C-4F). Finally, exposure to MβCD, a chemical that decreases cellular cholesterol content, also inhibited GBM cell survival (Fig. 4G). In conclusion, these results show that 24OHC suppresses GBM growth by depleting cellular cholesterol.

4: In Fig.6D, titration of efavirenz to concentrations exceeding 50µM has an increased effect in suppressing cell viability. However, according to Anderson et al. (JBC2016, cited by the authors) concentrations of 60µM and higher actually reduced CYP46A1 activity. While the data from Anderson are in vitro enzyme assays this still leaves the possibility that the effect of efavirenz seen by the authors might be an off-target effect. On way to assess this would be to downregulate or delete CYP46A1 in the cell lines by shRNA or CRISPR/Cas9. In those lines, efavirenz should have no effect. I feel this would be an experiment that could significantly strengthen the message of the manuscript that an already approved drug could be effective for treatment of GBM.

Response: This is a very good point. As suggested, we used siRNA constructs that were shown to robustly down-regulate CYP46A1 in GBM cells (LN229; Figure A below). We then used a cell proliferation assay to examine the effect of EFV on cells transfected with si-Ctrl or si-CYP46A1. The inhibitory effect of EFV (20 µM) was weaker in cells transfected with si-CYP46A1 compared to si-Ctrl ($P < 0.05$; Figure B below). These findings, together with our results in Fig. 6 and Fig. S9D, suggests that the anti-cancer effect of EFV (20 µM) in GBM is mediated through activation of

CYP46A1 function. We have now included these experiments in the revised manuscript, see Fig. S9E and S9F.

According to previous reports (Anderson et al, 2016), maximal CYP46A1 activation was observed at 20 μM EFV in an *in vitro* enzyme assay, with a subsequent decrease in the CYP46A1 stimulation and ultimately enzyme inhibition as the EFV concentrations were increased. In our present study, we observed a significant inhibition of GBM growth by EFV at concentrations exceeding 50 μM (Fig. 6D). Thus, EFV at higher concentrations was considered to be cytotoxic to cancer cells independent of its intended CYP46A1-activation properties.

EFV has been reported to exert anti-tumour effects in some other cancer types via several molecular mechanisms. For instance, in EFV-treated pancreatic cancer cells (over 40 μM), JNK and NF- κB signaling were significantly reduced after 24 h, which was accompanied by induction of oxidative stress and mitochondrial damage (Hecht et al, 2018). In breast cancer, mRNA microarray analysis revealed that EFV suppressed several well-characterized oncogenes, including epidermal growth factor receptor (EGFR) and the erythroblastic leukemia viral oncogene homolog (ERBB4), and upregulated genes involved in cell projection and the formation of dendritic spines (CDC42, FBXO2, EXOC4, RSHL3, TTLL9, and ATP1A2) (Patnala et al, 2014). Therefore, multi-omics studies such as RNA-seq and proteomics are needed to further investigate the molecular mechanisms underlying the effects of EFV at high concentrations in GBM cells.

5: The authors show that SREBP1 activity is not regulated on the level of its proteolytic cleavage but appears rather to be on the transcriptional level. Do the authors have any suggestion or potential candidate pathways for this regulation?

Response: SREBP1 is a transcription factor belonging to the basic helix-loop-helix leucine zipper family, and is considered to be involved in the transcriptional regulation of cholesterol-genic and lipogenic enzymes (Brown & Goldstein, 1997; Yokoyama et al, 1993). SREBP1 is synthesized as a precursor (P-SREBP1) bound to the endoplasmic reticulum and nuclear envelope. Upon activation, SREBP1 is cleaved, and the N-terminal end is released from the membrane into the nucleus as a mature protein (N-SREBP1) through a sequential two-step proteolytic process. N-SREBP1 binds to sterol regulatory elements (SRE) in the promoter regions of target genes. In western blot analysis, we found that both P-SREBP1 and N-SREBP1 expression levels were decreased in GBM cells under 24OHC treatment (Fig. 5H). These results indicate that 24OHC treatment significantly inhibits SREBP1 activity.

Recent studies have demonstrated a direct role of SREBP1 in maintaining its basal transcriptional activity (Takeuchi et al, 2010; Xu et al, 2016). An SRE site was identified in the promoter of *SREBP1*; when SREBP1 was knocked down, the promoter activity of the *SREBP1* gene was significantly reduced (Takeuchi et al, 2010), indicating that SREBP1 constitutes an autoloop regulatory circuit. Based on these findings, we hypothesize that the 24OHC-induced suppression of P-SREBP1 proteolytic processing might lead to reduced N-SREBP1 binding to the SRE site in its promoter region and therefore decreased P-SREBP1 transcription. However, further experiments are needed to clarify the precise mechanism of this autoloop regulatory circuit in GBM cells.

6: Materials and Methods: The authors should give more specific information on some of the reagents, such as catalog number/clone number when the description is ambiguous. I noted this especially for the antibodies used and reagents such as LDL (there are several types available from Millipore).

Response: We have carefully modified the Materials and Methods section (both in the main text as well as in supplementary M&M section) and included the catalog numbers and dilution used for each antibody.

References

Anderson KW, Mast N, Hudgens JW, Lin JB, Turko IV, Pikuleva IA (2016) Mapping of the Allosteric Site in Cholesterol Hydroxylase CYP46A1 for Efavirenz, a Drug That Stimulates Enzyme Activity. *J Biol Chem* 291: 11876-11886

Brown MS, Goldstein JL (1997) The SREBP pathway: regulation of cholesterol metabolism by proteolysis of a membrane-bound transcription factor. *Cell* 89: 331-340

- Gstraunthaler G (2003) Alternatives to the use of fetal bovine serum: serum-free cell culture. *ALTEX* 20: 275-281
- Guo D, Reinitz F, Youssef M, Hong C, Nathanson D, Akhavan D, Kuga D, Amzajerdi AN, Soto H, Zhu S et al (2011) An LXR agonist promotes glioblastoma cell death through inhibition of an EGFR/AKT/SREBP-1/LDLR-dependent pathway. *Cancer Discov* 1: 442-456
- Hecht M, Harrer T, Korber V, Sarpong EO, Moser F, Fiebig N, Schwegler M, Sturzl M, Fietkau R, Distel LV (2018) Cytotoxic effect of Efavirenz in BxPC-3 pancreatic cancer cells is based on oxidative stress and is synergistic with ionizing radiation. *Oncol Lett* 15: 1728-1736
- Keunen O, Johansson M, Oudin A, Sanzey M, Rahim SA, Fack F, Thorsen F, Taxt T, Bartos M, Jirik R et al (2011) Anti-VEGF treatment reduces blood supply and increases tumor cell invasion in glioblastoma. *Proc Natl Acad Sci U S A* 108: 3749-3754
- Patnala R, Lee SH, Dahlstrom JE, Ohms S, Chen L, Dheen ST, Rangasamy D (2014) Inhibition of LINE-1 retrotransposon-encoded reverse transcriptase modulates the expression of cell differentiation genes in breast cancer cells. *Breast Cancer Res Treat* 143: 239-253
- Pei S, Minhajuddin M, Adane B, Khan N, Stevens BM, Mack SC, Lai S, Rich JN, Inguva A, Shannon KM et al (2018) AMPK/FIS1-Mediated Mitophagy Is Required for Self-Renewal of Human AML Stem Cells. *Cell Stem Cell* 23: 86-100 e106
- Puchalski RB, Shah N, Miller J, Dalley R, Nomura SR, Yoon JG, Smith KA, Lankerovich M, Bertagnolli D, Bickley K et al (2018) An anatomic transcriptional atlas of human glioblastoma. *Science* 360: 660-663
- Takeuchi Y, Yahagi N, Izumida Y, Nishi M, Kubota M, Teraoka Y, Yamamoto T, Matsuzaka T, Nakagawa Y, Sekiya M et al (2010) Polyunsaturated fatty acids selectively suppress sterol regulatory element-binding protein-1 through proteolytic processing and autoloop regulatory circuit. *J Biol Chem* 285: 11681-11691
- Villa GR, Hulce JJ, Zanca C, Bi J, Ikegami S, Cahill GL, Gu Y, Lum KM, Masui K, Yang H et al (2016) An LXR-Cholesterol Axis Creates a Metabolic Co-Dependency for Brain Cancers. *Cancer Cell* 30: 683-693
- Xu HF, Luo J, Wang HP, Wang H, Zhang TY, Tian HB, Yao DW, Loo JJ (2016) Sterol regulatory element binding protein-1 (SREBP-1)c promoter: Characterization and transcriptional regulation by mature SREBP-1 and liver X receptor alpha in goat mammary epithelial cells. *J Dairy Sci* 99: 1595-1604
- Yokoyama C, Wang X, Briggs MR, Admon A, Wu J, Hua X, Goldstein JL, Brown MS (1993) SREBP-1, a basic-helix-loop-helix-leucine zipper protein that controls transcription of the low density lipoprotein receptor gene. *Cell* 75: 187-197

2nd Editorial Decision

10 October 2019

Thank you for the submission of your revised manuscript to EMBO Molecular Medicine. We have now received the enclosed reports from the referees that were asked to re-assess it. As you will see the reviewers are now globally supportive and I am pleased to inform you that we will be able to accept your manuscript pending minor editorial amendments.

Please submit your revised manuscript within two weeks. I look forward to seeing a revised form of your manuscript as soon as possible.

***** Reviewer's comments *****

Referee #2 (Comments on Novelty/Model System for Author):

This study is well designed and executed. The authors report a novel mechanistic connection between cholesterol metabolism, CYP46A1 and GBM growth. The translational impact of this manuscript is increased by in vivo experiments, where the authors successfully targeted GBM using Efavirenz.

Referee #2 (Remarks for Author):

The authors have addressed all my concerns.

Referee #3 (Remarks for Author):

The additional experimental data included significantly strengthen the manuscript and adequately address all questions I had. I now recommend this manuscript for publication and congratulate the authors for this interesting study.

2nd Revision - authors' response

19 October 2019

Authors made the requested editorial changes.

Corresponding Author Name: Rolf Bjerkvig
 Journal Submitted to: EMBO Molecular Medicine
 Manuscript Number: EMM-2019-10924